# Random sub-diffusion and capture of genes by the nuclear pore reduces dynamics and coordinates inter-chromosomal movement

Michael Chas Sumner[1], Steven B Torrisi[2], Donna G Brickner[1], Jason H Brickner[1]*

[1]Department of Molecular Biosciences, Northwestern University, Evanston, United States; [2]Department of Physics, Harvard University, Cambridge, United States

**Abstract** Hundreds of genes interact with the yeast nuclear pore complex (NPC), localizing at the nuclear periphery and clustering with co-regulated genes. Dynamic tracking of peripheral genes shows that they cycle on and off the NPC and that interaction with the NPC slows their sub-diffusive movement. Furthermore, NPC-dependent inter-chromosomal clustering leads to coordinated movement of pairs of loci separated by hundreds of nanometers. We developed fractional Brownian motion simulations for chromosomal loci in the nucleoplasm and interacting with NPCs. These simulations predict the rate and nature of random sub-diffusion during repositioning from nucleoplasm to periphery and match measurements from two different experimental models, arguing that recruitment to the nuclear periphery is due to random sub-diffusion and transient capture by NPCs. Finally, the simulations do not lead to inter-chromosomal clustering or coordinated movement, suggesting that interaction with the NPC is necessary, but not sufficient, to cause clustering.

*For correspondence:
Address correspondence to
j-brickner@northwestern.edu

Competing interests: The authors declare that no competing interests exist.

## Introduction

In eukaryotes, genomes are spatially organized within the nucleus. Chromosomes occupy distinct subnuclear 'territories', heterochromatin is segregated from euchromatin, and individual genes show non-random positioning relative to nuclear structures and other genes (*Misteli, 2020*). Gene positioning reflects physical interactions of chromosomal loci with nuclear structures like the nuclear lamina, nuclear pore complexes (NPCs), or nuclear bodies, and changes in gene expression are often accompanied by changes in gene positioning (*Brickner, 2017*). The positioning of genes can impact their transcription, mRNA processing, or chromatin modifications.

One model for such phenomena is the recruitment of genes to the nuclear periphery through interaction with the NPC. Many genes in budding yeast, *Caenorhabditis elegans*, *Drosophila*, and mammals physically interact with NPCs, suggesting that the NPC plays an important role in determining the spatial arrangement of eukaryotic genomes (*Brown et al., 2008a*; *Capelson et al., 2010*; *Casolari et al., 2004*; *Casolari et al., 2005*; *Ibarra et al., 2016*; *Jacinto et al., 2015*; *Liang et al., 2013*; *Pascual-Garcia et al., 2017*; *Rohner et al., 2013*; *Toda et al., 2017*). This is particularly apparent in budding yeast where hundreds of genes interact with the NPC and inducible genes rapidly reposition to the nuclear periphery upon activation (*Brickner and Walter, 2004*; *Casolari et al., 2005*; *Casolari et al., 2004*; *Van de Vosse et al., 2013*). Interaction with the NPC and localization to the nuclear periphery require specific transcription factors (TFs) and nuclear pore proteins (*Brickner et al., 2019*; *Brickner et al., 2012*; *Brickner et al., 2007*; *Cabal et al., 2006*; *Dieppois et al., 2006*; *Dilworth et al., 2005*; *D'Urso et al., 2016*; *Lapetina et al., 2017*; *Luthra et al., 2007*; *Randise-Hinchliff et al., 2016*; *Texari et al., 2013*; *Van de Vosse et al., 2013*).

A majority of yeast TFs can mediate interaction with the NPC (*Brickner et al., 2019*), suggesting that the yeast genome encodes spatial organization through *cis*-acting TF binding sites. Such *cis*-acting *DNA zip codes* are both necessary and sufficient to mediate interaction with the NPC and positioning to the nuclear periphery (*Ahmed et al., 2010*; *Brickner et al., 2019*; *Brickner et al., 2012*; *Light et al., 2010*; *Randise-Hinchliff et al., 2016*). Furthermore, interaction with the NPC frequently leads to inter-chromosomal clustering of co-regulated genes, suggesting that it influences the spatial organization of the yeast genome at multiple levels (*Brickner et al., 2016*; *Brickner et al., 2012*; *Kim et al., 2019*; *Kim et al., 2017*; *Mirkin et al., 2013*; *Randise-Hinchliff et al., 2016*).

Much of the work on gene recruitment to the nuclear periphery has utilized static population measurements such as microscopy, chromatin immunoprecipitation, or HiC. Although these studies have revealed important players necessary for gene positioning to the nuclear periphery, there are questions that cannot be answered using static methods. For example, while some loci interact very stably with the nuclear envelope (e.g., telomeres and centromeres; *Heun et al., 2001*; *Jin et al., 2000*), leading to ~85% of cells showing colocalization of these loci with the nuclear envelope, genes that interact with the NPC show lower levels (~50–65%; *Brickner and Walter, 2004*; *Casolari et al., 2004*). This has been suggested to reflect transient interaction with the nuclear periphery (*Brickner and Walter, 2004*), cell-cycle regulation of peripheral localization (*Brickner and Brickner, 2010*), or, perhaps, two distinct populations, one that stably associates with the NPC and the other that does not (*Brickner and Walter, 2004*; *Cabal et al., 2006*). Likewise, the repositioning of inducible genes from the nucleoplasm to the nuclear periphery is not well-understood. Some data – including the involvement of nuclear actin and myosin – has suggested that repositioning to the NPC could involve directed, super-diffusive movement (*Guet et al., 2015*; *Wang et al., 2020*). Finally, while inter-chromosomal clustering is a widespread phenomenon (*Apostolou and Thanos, 2008*; *Brickner et al., 2012*; *Brown et al., 2006*; *Homouz and Kudlicki, 2013*; *Kim et al., 2017*; *Lin et al., 2009*; *Noma et al., 2006*; *Schoenfelder et al., 2010*; *Thompson et al., 2003*), relatively few studies have explored the dynamics of clustering over time and it is unclear if clustering reflects a stable physical interaction (*Brickner et al., 2016*; *Dai et al., 2018*). High-resolution, quantitative dynamics of chromatin diffusion are required to address each of these questions.

Chromatin is a mobile polymer, and individual loci exhibit constrained or anomalous diffusion (*Bystricky et al., 2004*; *Gasser, 2002*; *Hajjoul et al., 2013*; *Heun et al., 2001*; *Marshall et al., 1997*). Chromatin motion can reveal important aspects of the nuclear environment and the biophysical mechanisms that control the spatial organization of the genome. Repositioning to the NPC in budding yeast is an intriguing model for such studies because it is inducible, relatively rapid, controlled by well-understood DNA elements, and induces both a change in position and inter-chromosomal clustering.

Here we show that repositioning to the nuclear periphery is continuous and dynamic but uniform within the population, suggesting that, within each cell, localization to the periphery it is a probabilistic process. Localization at the nuclear periphery correlates with more constrained diffusion, as suggested by previous work (*Backlund et al., 2014*; *Cabal et al., 2006*). Using mean-squared displacement (MSD) analysis and molecular genetics, we pinpoint this effect to the interaction with the NPC. The parameters of sub-diffusion derived from MSD of nucleoplasmic loci were used to develop a computational simulation that faithfully recapitulates the behavior of such genes. This simulation was also adapted to model repositioning to the nuclear periphery through random sub-diffusion and transient capture at the nuclear envelope. The repositioning predicted by the simulation was then compared with several rapid repositioning experiments to determine whether it is vectorial or super-diffusive. The simulation matched the observed behavior of loci in cells, suggesting that repositioning from the nucleoplasm to the nuclear periphery does not require directed movement.

Finally, we monitored the dynamics of inter-chromosomal clustering. Unlike pairs of simulated paths, genes that exhibit clustering remain near each other for tens of seconds and show correlated movement. Simulated interaction with the NPC, while sufficient to recapitulate the chromatin dynamics of individual loci, is not sufficient to recapitulate this correlated movement. Therefore, we propose that inter-chromosomal clustering relies on a distinct physical interaction between genes that can extend hundreds of nanometers.

# Results

## Chromatin positioning to the nuclear periphery is continuous and dynamic

The localization of genes at the nuclear periphery can be followed in live yeast cells by tagging chromosomal loci of interest with an array of 128 Lac operators in a strain expressing GFP-Lac repressor (GFP-LacI) and quantifying its colocalization with mCherry-marked nuclear envelope (*Figure 1A*; *Brickner and Walter, 2004*; *Egecioglu et al., 2014*; *Robinett et al., 1996*; *Straight et al., 1996*). In static confocal microscopy experiments, repositioning of inducible genes such as *HIS4* or *INO1* to

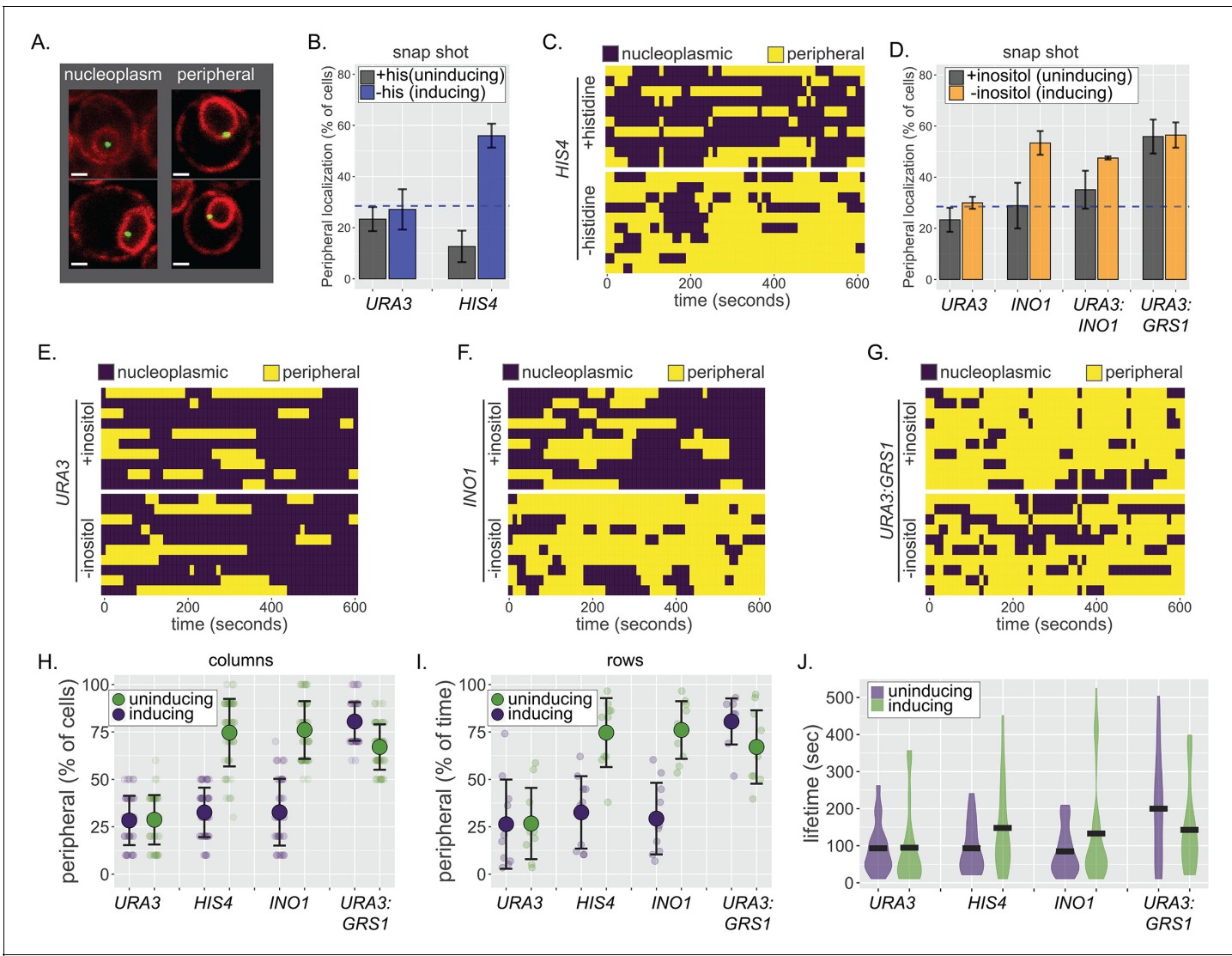

**Figure 1.** Continuous and dynamic positioning at the nuclear periphery. (A) Representative confocal micrographs of cells having the LacO array integrated at a locus of interest, expressing GFP-LacI and Pho88-mCherry (*Robinett et al., 1996*; *Brickner and Walter, 2004*; *Brickner et al., 2019*) and scored as either nucleoplasmic (left) or peripheral (right). (B) Peripheral localization (% of cells ± SEM) of *URA3* and *HIS4* in cells grown ± histidine. The hatched blue line, here and throughout: peripheral localization predicted by chance. (C, E–G) Kymographs of 10 cells with a LacO array integrated at *HIS4* (C), *URA3* (E), *INO1* (F), or *URA3:GRS1* (G) were grown in the indicated medium and scored for peripheral localization every 10 s for 5 min. Yellow: peripheral; purple: nucleoplasmic. (D) Peripheral localization (± SEM) of *URA3*, *INO1*, *URA3:INO1*, and *URA3:GRS1* in cells grown ± inositol. (H–J) Summary plots from (C, E–G): (H) mean percentage of cells (± SD) in which the locus is peripheral at each time point (i.e., each dot represents a summary of a single column from kymographs); (I) mean percentage of time (± SD) each locus spent colocalized with the nuclear envelope (i.e., each dot represents a summary of a single row from kymographs); and (J) the distribution and median duration of periods of peripheral localization of each locus.

the periphery leads to an increase in the fraction of cells in which the locus colocalizes with the nuclear envelope from that expected for a random distribution (~30%) to ~50–65% (*Figure 1B,D*; *Brickner and Walter, 2004*; *Egecioglu et al., 2014*). However, artificially tethering chromatin to the nuclear envelope leads to ~85% colocalization with the nuclear envelope (*Brickner and Walter, 2004*). This suggests that localization to the nuclear periphery reflects either dynamic or continuous interaction with the NPC or two distinct populations of cells, one that exhibits stable association with the nuclear envelope and the other that does not. To distinguish between these possibilities, we quantified peripheral localization of three LacO-tagged loci over time in individual cells: the inducible genes *HIS4* and *INO1*, as well as the negative control *URA3*, which localizes in the nucleoplasm (*Figure 1B,D*; *Brickner et al., 2019*; *Brickner and Walter, 2004*; *Randise-Hinchliff et al., 2016*). To avoid the complication that interaction of many genes with the NPC is lost during S-phase (*Brickner and Brickner, 2010*), cells were synchronized using nocodazole and released into G1 for 30 min before scoring colocalization with the nuclear envelope every 10 s over 10 min. In complete media (i.e., uninducing conditions), all three genes showed similar patterns: episodic, brief colocalization with the nuclear envelope (*Figure 1C, E, and F*). However, under inducing conditions (−histidine for *HIS4* or −inositol for *INO1*), the pattern changed. Both *HIS4* and *INO1* showed longer periods of colocalization with the nuclear envelope (*Figure 1C, F, and J*), while *URA3* was unaffected (*Figure 1E*). The pattern was consistent across the population, so that the fraction of cells in which *HIS4* or *INO1* colocalized with the nuclear envelope at each time point (*Figure 1H*) was in close agreement with the fraction of time spent colocalized with the nuclear envelope in each cell (*Figure 1I*). This argues against two distinct populations and instead suggests that interaction with the NPC is continuous and dynamic over time, increasing the duration of colocalization with the nuclear envelope.

Interaction with the NPC is mediated by TFs binding to *cis*-acting elements that function as *DNA zip codes* (*Ahmed et al., 2010*; *Brickner et al., 2019*; *Light et al., 2010*). For example, the Gene Recruitment Sequence GRS1 from the *INO1* promoter binds to the Put3 TF to mediate interaction with the NPC and positioning at the nuclear periphery (*Brickner et al., 2012*). Likewise, the Gcn4 binding site (GCN4 BS) from the *HIS3* promoter is sufficient to mediate interaction with the NPC (*Randise-Hinchliff et al., 2016*). Inserting zip codes near *URA3* is sufficient to reposition *URA3* to the nuclear periphery (e.g., *URA3:GRS1*, *Figure 1D*; *Ahmed et al., 2010*; *Randise-Hinchliff et al., 2016*). The association of *URA3:GRS1*, which shows unregulated localization to the periphery, with the nuclear envelope over time resembled that of active *HIS4* and *INO1* (*Figure 1G–J*). Thus, DNA zip code-mediated interaction with the NPC is sufficient to produce continuous and dynamic association with the nuclear envelope.

## Chromatin sub-diffusion is suppressed by interaction with the NPC

We next examined how interaction of genes with the NPC impacts the dynamics of diffusion using MSD analysis. MSD has been used to show that chromosomal loci exhibit constrained sub-diffusion (*Marshall et al., 1997*). For comparison, we tracked the movement of the less-mobile nuclear envelope-embedded spindle pole body (SPB) and a much more mobile cytoplasmic particle (the μNS viral capsid; *Munder et al., 2016*). While μNS was highly diffusive, the SPB showed very limited displacement at this timescale, reflecting both slow diffusion within the membrane and movement of the whole nucleus (*Figure 2B*). The MSD of 11 nucleoplasmic loci (i.e., not associated with the NPC) and two telomeres tethered to the nuclear envelope exhibited a range of intermediate sub-diffusion between these two extremes, with the nucleoplasmic loci showing greater MSD than tethered telomeres and telomeres showing greater MSD than the SPB (*Figure 2B*; *Supplementary file 1*). Simultaneously acquiring images of chromosomal loci and the SPB to correct for nuclear movement significantly reduced the time resolution (data not shown). Given that nuclear movement was much less than chromosomal movement at these timescales, it could be ignored. We also determined the MSD of chromosomal loci in 3D. Although this gave very similar results (*Supplementary file 1*), the quality of the data was lower because of the longer time interval (>1 s). For these reasons, we limited our movies for MSD analysis to 40 s at 210 ms resolution (200 × 0.21 s) in a single focal plane and calculated MSD for time intervals between 210 ms and 4 s (*Figure 2B*).

The nucleoplasmic loci showed a range of mobility by MSD, perhaps reflecting nearby physical interactions with the nuclear envelope. Tethering to the nuclear envelope has a significant effect on chromatin positioning and the fraction of the nuclear volume explored over distances below 30 kb

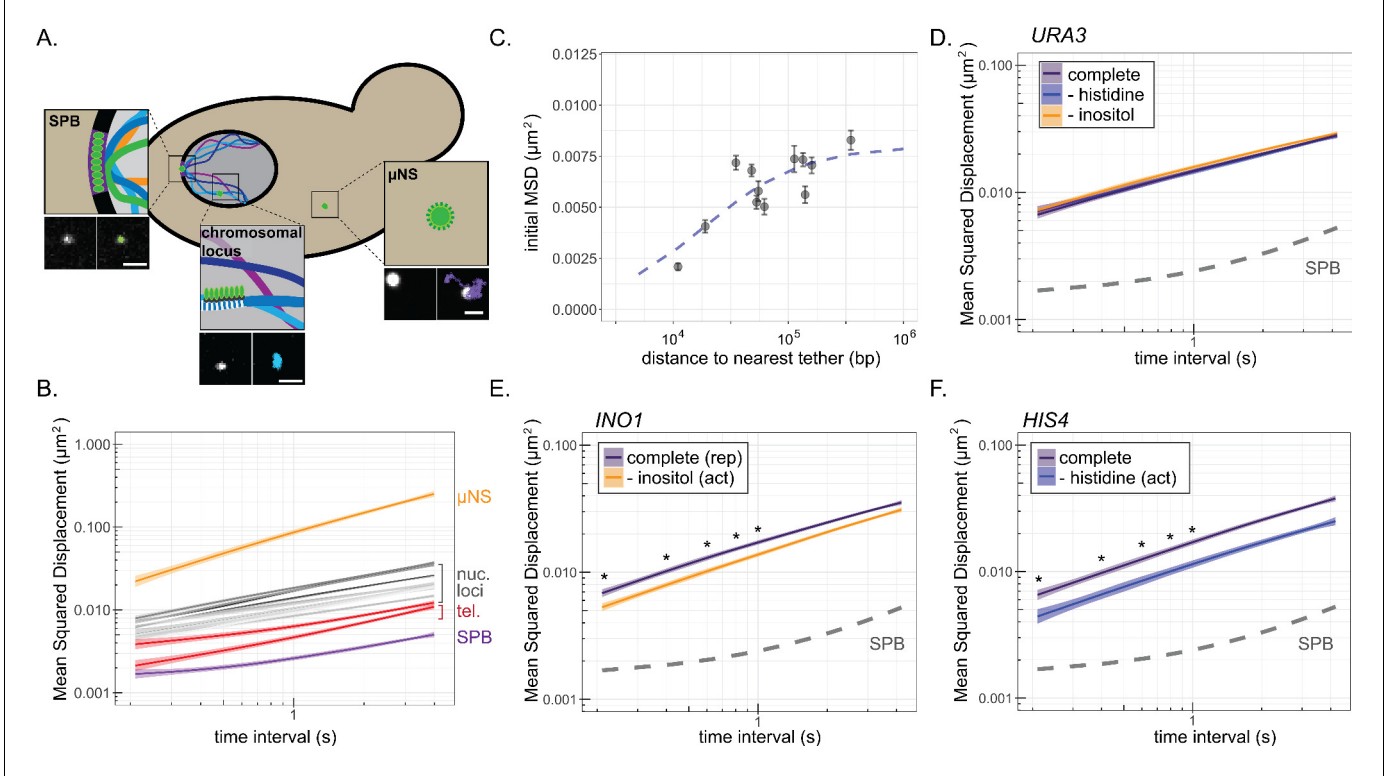

**Figure 2.** Mean-squared displacement (MSD) of chromatin sub-diffusion. (**A**) Schematic of fluorescent foci within the yeast cell. Fluorescently tagged spindle pole body (SPB), cytoplasmic μNS, and chromosomal locus were tracked over 200 × 200 ms. Example micrographs of each particle (left) and overlaid path (right) are shown for each. Scale bar = 1 μm. (**B**) Average MSD for μNS (orange), SPB (purple), 10 nucleoplasmic loci (gray; listed in **Supplementary file 1**) and two telomeres (red) at different time intervals (τ). The ribbon around the mean represents standard error. (**C**) Mean MSD ± standard deviation for τ = 200 ms for each chromosomal locus in (**B**) vs log₁₀ (base pairs) to the nearest tether point (centromere or telomere). The line is from the fit of the data to a non-linear model for a hyperbolic curve, as described in the text. (**D–F**) MSD plots of *INO1* (**D**), *URA3* (**E**), or *HIS4* (**F**) in cells grown in the indicated media. In all plots, the dashed line represents the MSD of the SPB. *p<0.05 based on Kolmogorov–Smirnov test comparing MSDs at the indicated times.

The online version of this article includes the following source data and figure supplement(s) for figure 2:

**Source data 1.** Comma-separated tables of tracking data used for *Figure 2B, D, E, and F*.

**Figure supplement 1.** MSD of peripheral vs nucleoplasmic cells.

(**Avşaroğlu et al., 2014**; **Verdaasdonk et al., 2013**). Indeed, the initial MSDs (τ = 0.21 s) showed a non-linear relationship to the genomic distance to the nearest nuclear envelope tethering point (either centromeres or telomeres; *Figure 2C*). Consistent with work from others, we could model this relationship as a hyperbolic curve with a half-maximal MSD observed at ~18 kb (*Figure 2C*, blue dashed line; **Avşaroğlu et al., 2014**; **Verdaasdonk et al., 2013**). Thus, diffusion of chromatin is influenced over relatively short distances by stable interactions with the nuclear envelope (**Hediger et al., 2006**; **Hediger and Gasser, 2002**).

To quantify the effect of local interaction with the NPC on chromatin sub-diffusion, we examined genes that show conditional association with the NPC. We compared the MSD of *INO1*, *HIS4*, and *URA3* under either uninducing or inducing conditions (±histidine and ±inositol). As expected, *URA3* showed no change in MSD under these conditions (*Figure 2D*). However, both *HIS4* and *INO1* showed significantly reduced mobility upon induction (*Figure 2E,F*), confirming that repositioning to the nuclear periphery correlates with reduced chromatin sub-diffusion.

To further strengthen this correlation, we exploited the population dynamics illuminated in *Figure 1*, performing MSD analysis on sub-populations of cells in which the locus was either stably maintained at the nuclear periphery (i.e., those cells in which >50% of the time points were peripheral) or predominantly in the nucleoplasm (<10% peripheral) during the 40 s acquisition (*Figure 2—figure supplement 1*). When we performed this analysis with repressed *INO1*, the MSD from

predominantly peripheral cells was indistinguishable from the MSD from predominantly nucleoplasmic cells (*Figure 2—figure supplement 1C*). However, for active *INO1*, the MSD from predominantly peripheral cells was significantly lower than the MSD from predominantly nucleoplasmic cells (*Figure 2—figure supplement 1D*), consistent with the decrease in MSD resulting from interaction with the NPC.

If the change in MSD is due to interaction with the NPC, a DNA zip code integrated at an ectopic site should also reduce MSD. Single copies of zip codes from the promoters of *INO1* (*URA3:GRS1*; *Figure 3A*) or *HIS4* (*URA3:GCN4BS*; *Figure 3B*) were integrated at the *URA3* locus. *URA3:GRS1* localizes at the nuclear periphery constitutively (*Figures 1D* and *3A*; *Ahmed et al., 2010*; *Randise-Hinchliff et al., 2016*), resulting in a reduced MSD under all conditions. In contrast, *URA3:GCN4BS* shows conditional localization to the periphery upon amino acid starvation (*Figure 3B*, inset; *Randise-Hinchliff et al., 2016*), and a conditional reduction in MSD (*Figure 3B*). Loss of the NPC protein Nup2 disrupts DNA zip code-mediated localization to the nuclear periphery and resulted in MSD similar to *URA3* under all conditions (*Figure 3C,D*). Thus, DNA zip code-mediated interaction with the NPC is sufficient to suppress chromatin sub-diffusion.

Transcriptional activation and chromatin remodeling can cause increased chromatin mobility (*Gasser et al., 2004*; *Gu et al., 2018*). Therefore, to disentangle the effects of peripheral localization from the effects of transcriptional activity on MSD, we monitored MSD in mutants that lack *trans*-acting transcriptional regulators of the *INO1* gene. Both *INO1* transcription and *INO1* interaction with the NPC are regulated by the Opi1 repressor, which recruits the Rpd3L histone deacetylase to regulate binding of the Put3 TF to the GRS1 zip code (*Randise-Hinchliff et al., 2016*). Because Opi1 is recruited to the *INO1* promoter by binding to the Ino2 activator (*Heyken et al., 2005*), loss of either Ino2 or Opi1 leads to constitutive peripheral localization (*Figure 3E,F*, insets; *Randise-Hinchliff et al., 2016*). However, these two mutants have opposite effects on *INO1* transcription: *ino2Δ* blocks all expression, while *opi1Δ* shows unregulated, high-level expression (*Greenberg et al., 1982a*; *Greenberg et al., 1982b*). In both mutants, the *INO1* MSD resembled that of active *INO1* (*Figure 3E,F*), suggesting that interaction with the NPC is the principal cause of the decrease in sub-diffusion.

## Simulating chromatin sub-diffusion and repositioning to the nuclear periphery

Using parameters from the MSD analysis, we developed a simulation of chromatin sub-diffusion (https://github.com/MCnu/YGRW). Sub-diffusion of a segment of chromatin results from forces affecting the chromatin segment both directly (e.g., the viscoelastic potential of the polymer, boundary collision) and indirectly (forces and membrane tethering nearby; *Figure 2C*). MSD for a Rouse polymer like chromatin reflects a relationship $MSD(\tau) = \Gamma(\tau\alpha)$ for any time interval $\tau$ (*Socol et al., 2019*). Gamma ($\Gamma$) describes the diffusion coefficient, while an $\alpha$ exponent less than one reflects a hallmark for sub-diffusive movement: each step vector is anticorrelated with both the previous and subsequent steps (*Lucas et al., 2014*). While the exact value for $\alpha$ from different MSD experiments or different loci varies (*Backlund et al., 2014*), work on multiple loci in yeast (*Hajjoul et al., 2013*) and our MSD data with nucleoplasmic loci (see Materials and methods) suggests that yeast chromatin has an average $\alpha = 0.52$.

Chromatin sub-diffusion has been modeled using several approaches (*Arbona et al., 2017*; *Verdaasdonk et al., 2013*). Anticorrelated movement cannot be reproduced through either a random walk or a simple process of weighted step sizes derived from our experimental observations (*Figure 4—figure supplement 1A,B*; uniform and Gaussian, respectively). However, a continuous-time Gaussian process known as fractional Brownian motion (FBM) produces trajectories that approximate chromatin sub-diffusion (*Lucas et al., 2014*). FBM produces non-independent steps across time, allowing us to impart the anticorrelation between individual steps that is characteristic of yeast chromatin sub-diffusion. For each trajectory, two numeric arrays for the x and y dimensions of movement (*Dietrich and Newsam, 1997*) were generated based on an expected covariance matrix and $\alpha = 0.52$. This array produces a stochastic time series of vectors with an anticorrelation structure functionally identical to that observed for chromatin movement. Finally, these vectors were scaled according to the experimentally derived $\Gamma$ value and Hurst exponent ($\alpha/2$; *Mandelbrot and Van Ness, 1968*). Starting from random positions within the nucleus, the resulting array of discrete step lengths describes a single, two-dimensional sub-diffusive particle trajectory. This simple and

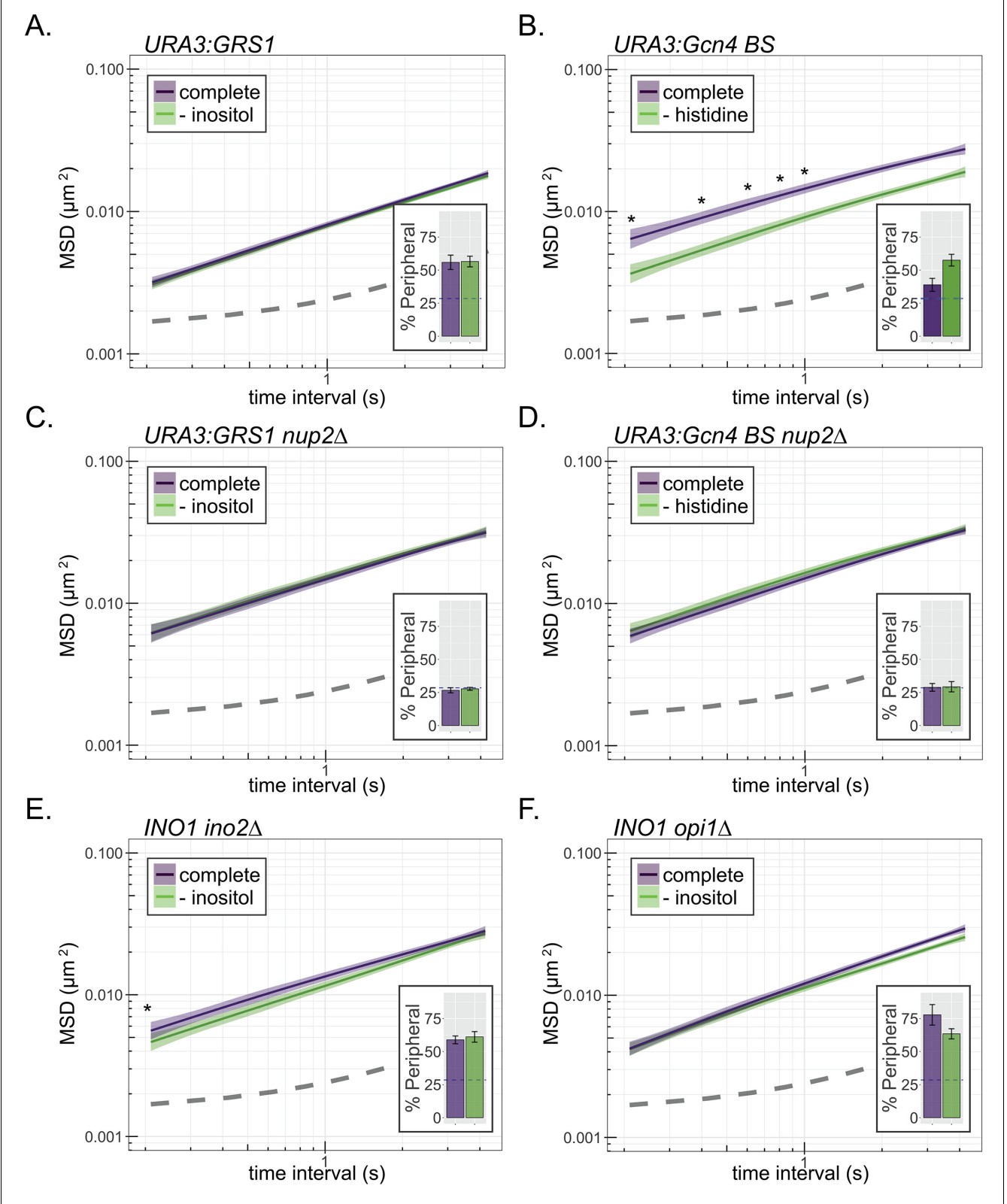

**Figure 3.** Interaction with the NPC reduces chromatin sub-diffusion. (A–F) MSD of *URA3* (A–D) and *INO1* (E, F) in strains grown in the indicated media. Dark line indicates average MSD, ribbon = bootstrapped SEM. Insets: peripheral localization of each locus (mean % of cells ± SEM). The GRS1 zip code from the *INO1* promoter (A, C) or the Gcn4 binding site (B, D) was integrated and integrated at *URA3* in wild-type (A, B) or *nup2Δ* (C, D) strains. MSD of *INO1* in *ino2Δ* (E) or *opi1Δ* (F) strains. *p<0.05 based on Kolmogorov–Smirnov test comparing MSD at the indicated time points.

*Figure 3 continued on next page*

eLife Research article

Cell Biology | Chromosomes and Gene Expression

rapid approach generates trajectories similar to our experimental observations and imparts memory resembling the MSD of chromosomal loci in the nucleoplasm (*Figure 4—figure supplement 1A and B*).

Paths generated by FBM suffer from one significant shortcoming. In an enclosed volume, FBM will deplete occupancy of particles near the boundary over time, resulting in a biased distribution (*Figure 4—figure supplement 1C*). This phenomenon has also been reported by others (*Vojta et al., 2020*) and is not consistent with observations that chromosomal loci, unless associated with the nuclear envelope, localize at the nuclear periphery at a frequency expected from a random distribution (*Brickner and Walter, 2004*; *Hediger et al., 2002*). This may reflect a fundamental difference between sub-diffusion of particles and the apparent sub-diffusion of a segment of chromatin. We explored several methods to avoid depletion at the nuclear periphery and found that the following was effective: steps that would have taken the locus beyond the boundary were replaced with steps to the boundary along the same vector and, upon interaction with the boundary, the normalized, correlated noise for future steps was regenerated (*Figure 4A, Figure 4—figure supplement 1*, FBM + regeneration). This modified simulation produced paths that closely matched the MSD, the distribution of positions within the nucleus, and the peripheral occupancy of nucleoplasmic chromosomal loci (*Figure 4B,E–G*; loci within 150 nm of the membrane in the simulation were scored as peripheral).

From our model for nucleoplasmic gene movement, we sought to simulate chromatin interaction with NPCs at the nuclear membrane. Based on the height of the NPC basket (*Yang et al., 1998*; *Vallotton et al., 2019*), we created a zone 50 nm from the boundary where chromatin could become 'bound', causing it to switch to SPB-like sub-diffusion (*Figure 2B*). The probabilities of binding and unbinding within this zone were varied independently to optimize the agreement with the experimental MSD and peripheral localization (i.e., localization within 150 nm of the nuclear envelope) of *URA3:GRS1* (*Figure 4—figure supplement 2*). Based on this optimization, we found that a binding probability of 0.9 and a probability of remaining bound of 0.95 resulted in a positional distribution (*Figure 4D*), peripheral occupancy over time (*Figure 4E,F*), and MSD (*Figure 4G*) that most closely matched that of *URA3:GRS1*. We refer to this modified simulation as simulation+zip code. The fit of the simulation to the mean MSD for *URA3* and of the simulation+zip code to the mean MSD for *URA3:GRS1* was excellent (Pearson's $X^2$ sums of 0.001 and 0.003, respectively, for $\tau$ from 0.21 to 4 s). Together, these two relatively simple simulations capture important aspects of chromatin sub-diffusion and gene positioning at the nuclear periphery.

## Chromatin repositioning is achieved by random sub-diffusion and capture

Chromosomal loci can undergo long-range, directed movement (*Miné-Hattab and Rothstein, 2013*), raising the possibility that repositioning from the nucleoplasm to the nuclear periphery could be an active process. Furthermore, actin and the myosin motor Myo3 have been shown to play a role in the localization of *INO1* to the nuclear periphery (*Wang et al., 2020*). We find that deletion of Myo3 leads to a delay in the targeting of *URA3:INO1* to the nuclear periphery (*Figure 5—figure supplement 1A & B*). Importantly, this defect is specific to one (GRS1) of the two DNA zip codes that mediate repositioning of *INO1* to the nuclear periphery (*Ahmed et al., 2010*). When both zip codes (GRS1 and GRS2) are present at the endogenous *INO1* gene, loss of Myo3 had no effect (not shown). Furthermore, once positioned at the nuclear periphery, *URA3:INO1* localization was unaffected by degradation of Myo3-AID (auxin-inducible degron; *Figure 5—figure supplement 1C*), suggesting that Myo3 increases the rate or efficiency of repositioning to the nuclear periphery. The MSD of *URA3:INO1* in the *myo3Δ* mutant reflected its localization; under repressing conditions or after only 1 hr of inositol starvation, the MSD was unchanged, whereas after 24 hr of inositol starvation, MSD decreased (*Figure 5—figure supplement 1D*). These results suggest that Myo3 is

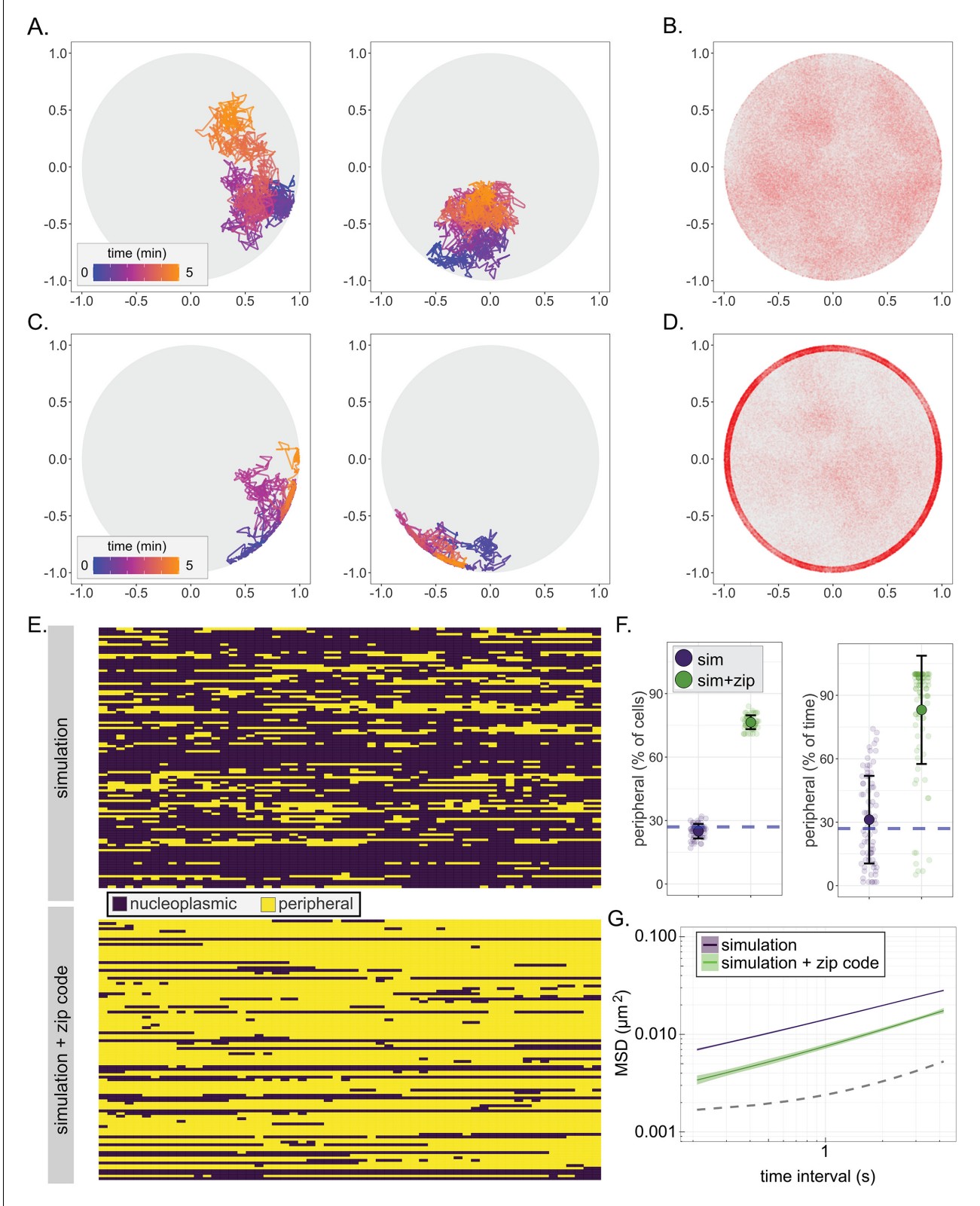

**Figure 4.** A fractional Brownian motion simulation of chromatin sub-diffusion. (**A, C**) Randomly selected example paths over 5 min at 200 ms time resolution. Color scale represents time. Paths were simulated using parameters (diffusion coefficient and anomalous exponent) extracted from a non-linear regression fit to *URA3* MSD (**A**; simulation) or by also allowing interaction at the nuclear envelope, slowing sub-diffusion to that of the SPB (**C**; simulation+zip code). (**B, D**) 150,000 positions visited in 100 simulated 5 min paths at 200 ms time resolution for the simulation (**B**) or the simulation+zip

*Figure 4 continued on next page*

*Figure 4 continued*

code (D). (E) Peripheral localization (i.e., positioned ≤150 nm from the edge of the nucleus) every 10 s over 10 min for 100 paths from the simulation (top) and simulation+zip code (bottom). (F) Summary plots for percent of cells in that scored as peripheral at each time (left) or the percent of time each cell scored as peripheral (right) in either the simulation or the simulation+zip code. (G) MSD of the paths from the simulation or the simulation+zip code. Dark line is the mean, and the colored band represents the bootstrapped standard error.

The online version of this article includes the following figure supplement(s) for figure 4:

**Figure supplement 1.** Comparison of simulations of sub-diffusion of nucleoplasmic chromatin.

**Figure supplement 2.** Optimization of binding probability and retention probability by comparison with *URA3:GRS1*.

impacting either the movement of *URA3:INO1* from the nucleoplasm to the nuclear periphery other regulatory steps that are necessary for rapid GRS1-mediated peripheral localization.

To explore whether directed movement is responsible for repositioning of genes from the nucleoplasm to the nuclear periphery, we first determined the behavior of the simulation, which does not possess active, vectorial movement. Initiating either the default simulation of chromatin movement or the simulation+zip code from random positions within the nucleus, we followed the percent of the population showing localization within 150 nm of the nuclear edge over time. For the nucleoplasmic simulation, the peripheral localization remained random over time (~28% peripheral; *Figure 5A*). However, interaction with the nuclear envelope in the simulation+zip code resulted in stable repositioning to the nuclear periphery within ~2 min (*Figure 5A*). Therefore, rapid repositioning to the nuclear periphery can occur without any directed, active movement.

To compare these simulations with experimental results, we applied live-cell tracking during repositioning from the nucleoplasm to the periphery. One challenge with such experiments is that the time required for genes to reposition when cells are shifted from uninducing to inducing conditions is gene-specific and can be quite slow (e.g., $t_{1/2}$ ~ 30min; *Brickner et al., 2012*, *Brickner et al., 2007*; *Randise-Hinchliff et al., 2016*). This suggests that the rate-limiting step for repositioning often reflects the regulation of TFs that mediate repositioning, rather than the rate-limiting step for movement to the periphery (*Randise-Hinchliff et al., 2016*). To overcome this complication, we developed two approaches to maximize the rate of repositioning from the nucleoplasm to the nuclear periphery. First, we arrested cells bearing *URA3:GRS1-LacO* with α-factor mating pheromone, which disrupts peripheral localization by inhibiting Cdk, which phosphorylates Nup1 and is required for peripheral localization of *URA3:GRS1* (*Brickner and Brickner, 2010*). Upon release from α-factor arrest, *URA3:GRS1* repositioned to the nuclear periphery within ~15 min (*Figure 5C*).

Tethering of a 27 amino acid 'positioning domain' from the Gcn4 TF ($PD_{GCN4}$) near *URA3* using the LexA DNA binding domain (DBD) is sufficient to position *URA3:LexABS* at the nuclear periphery (*Brickner et al., 2019*). Therefore, as a complementary approach, we used an optogenetic switch to recruit the $PD_{GCN4}$ to *URA3*, resulting in targeting to the nuclear periphery. Cryptochrome 2 (CRY2) and cryptochrome interacting protein CIB1 from *Arabidopsis thaliana* undergo rapid dimerization when exposed to 488 nm light (*Benedetti et al., 2018*). In a strain having both the LacO array and the LexA binding site at *URA3*, CRY2-LexA DBD was co-expressed with CIB1-$PD_{GCN4}$ to generate a light-induced peripheral localization system (*Figure 5D*; *Brickner et al., 2019*). LexA DBD-Gcn4 served as a positive control and a mutant CIB1-$pd_{GCN4}$ that does not mediate interaction with the NPC served as a negative control (*Brickner et al., 2019*). Cells were arrested, synchronized in G1, and illuminated with 488 nm light for 1 s pulses every 10 s over 10 min. Illumination resulted in rapid, $PD_{GCN4}$-dependent repositioning to the nuclear periphery within ~7.5 min (*Figure 5E*). Thus, both the biological and the optogenetic stimuli led to rapid repositioning to the nuclear periphery with kinetics comparable to the simulation.

Having established that these two approaches lead to rapid peripheral localization, we then used particle tracking to define the nature of the movement during this transition. *URA3*, *URA3:GRS1*, or *URA3:LexABS* were tracked for 5 min at 0.5 s resolution (600 frames) during repositioning. For each movie, the position and time of initial colocalization with the nuclear envelope was recorded (if observed). While peripheral colocalization of *URA3:GRS1* and *URA3:LexABS*+CIB1-$PD_{GCN4}$ represents – at least some of the time – interaction with the NPC, peripheral colocalization of the negative controls does not. Therefore, we expected that if directed movement brings genes to the nuclear periphery, the positive and negative controls should show differences in the step velocities, time of

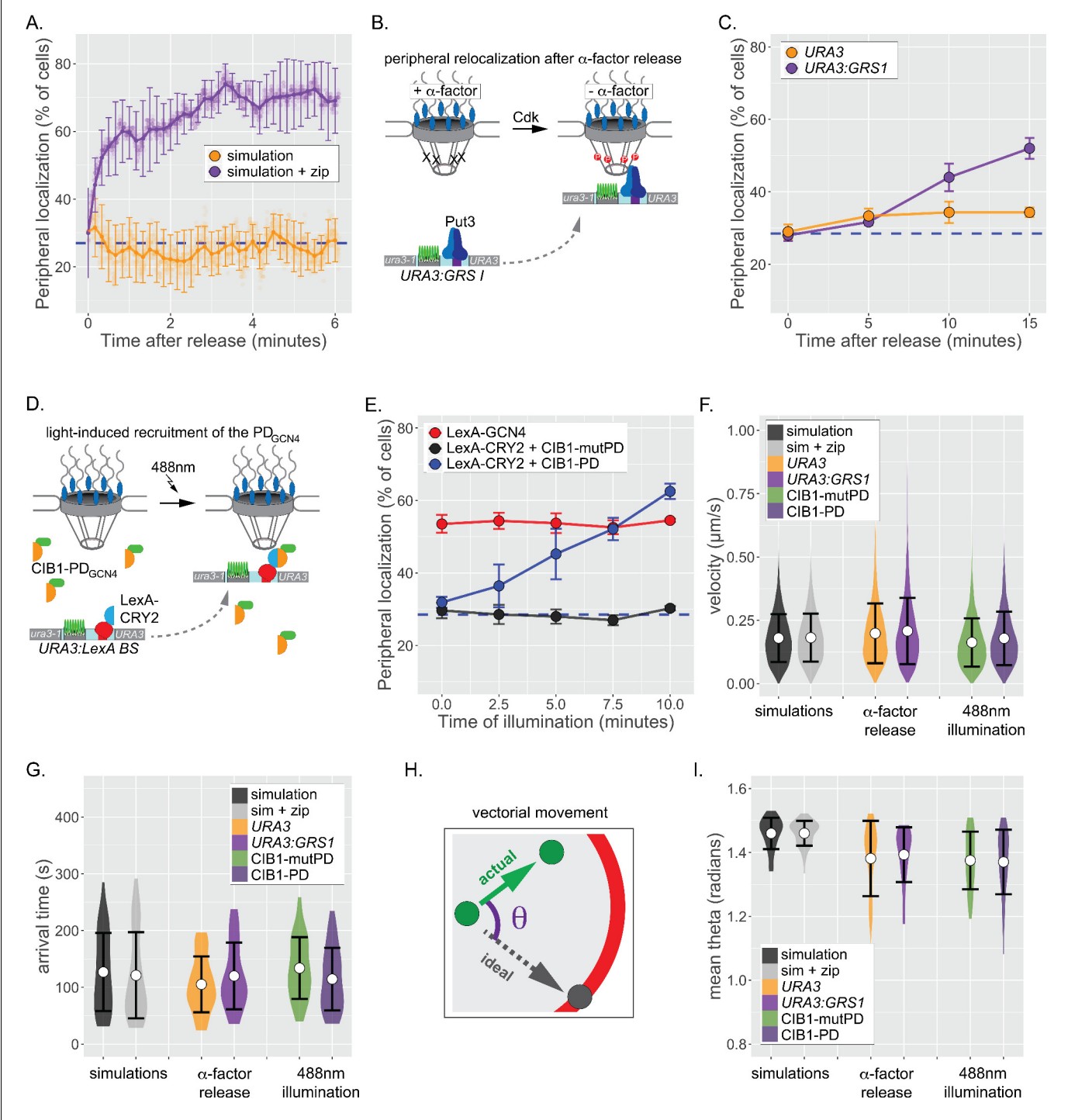

**Figure 5.** Repositioning from the nucleoplasm to the NPC. (**A**) Simulated repositioning. Simulated paths, using either the fractional Brownian simulation or the simulation+zip code, were initiated at random positions within 2 μm diameter nucleus and followed for 20 min (200 ms resolution). Colocalization with the periphery (i.e., ≤150 nm from the edge) was scored for each simulation at each time and smoothed by averaging over 10 s windows. For each time point, three replicates of 33 paths were scored to generate an average (points) ± SEM (error bars). Blue, hatched line: peripheral localization expected for a random distribution. (**B**) Schematic for repositioning to the nuclear periphery upon release from α-factor arrest. (**C**) Peripheral localization (% of cells ± SEM) of *URA3* or *URA3:GRS1* over time after removing α-factor. (**D**) Schematic for optogenetic light-induced repositioning to the nuclear periphery. (**E**) Peripheral of *URA3:LexABS* in strains expressing either LexA-GCN4, LexA-CRY2+mutant PD$_{GCN4}$-CIB1, or LexA-CRY2+wild-type PD$_{GCN4}$ at the indicated times after illumination with 488 nm light. (**F–I**) Summary plots of velocity (**F**), arrival time (**G**), and angular deviation from an ideal path (**I**) from each cell before initial colocalization with nuclear periphery. White circles are the mean values, and error bars represent the
*Figure 5 continued on next page*

*Figure 5 continued*

standard deviation. For (F–I), simulated paths were initiated at random positions within a 1 μm diameter sphere in the center of the 2 μm diameter nucleus and followed for 5 min. Paths that did not make contact with the nuclear periphery were excluded.

The online version of this article includes the following source data and figure supplement(s) for figure 5:

**Source data 1.** Comma-separated tables of simulated paths and tracking data used for *Figure 5*.
**Figure supplement 1.** Loss of Myo3 delays GRS1-dependent repositioning to the nuclear periphery.

arrival, or directness of the path preceding arrival at the nuclear periphery. For comparison, we also determined each of these parameters for paths generated by the default simulation and the simulation+zip code, which include no directed movement. The mean velocities for the simulations and experimental controls were statistically indistinguishable, ranging from $0.163 \pm 0.10$ μm s$^{-1}$ to $0.207 \pm 0.13$ μm s$^{-1}$ (*Figure 5F*; n = 6077–9724 steps per strain), suggesting that the speed of movement was not increased during peripheral repositioning. We did not observe significantly more large steps in the experimental movies than in the negative control movies (*Figure 5F*). The mean arrival time prior to initial contact with the nuclear envelope was also similar between the simulations and the experimental controls, ranging from $105 \pm 49$ s to $133 \pm 54$ s (*Figure 5G*; n = 27–40 cells per strain), consistent with the predictions from the simulation. Finally, to assess whether any of the loci underwent processive, vectorial movement during translocation, we measured the radial deviation (θ) of each step from a direct path to the ultimate contact point at the nuclear envelope (*Figure 5H*). Random sub-diffusion should produce an average θ of $\sim \pi/2 = 1.57$ radians, while directed movement would produce an average of ~0. The simulations were close to random, and while the experimental loci appear slightly more directed than random, the positive and negative controls were indistinguishable (*Figure 5I*). Taken together, these results indicate that repositioning of chromatin from the nucleoplasm to the nuclear periphery is likely due to random sub-diffusion and collision with the NPC.

## Dynamics of inter-chromosomal clustering

Genes that interact with the yeast NPC can exhibit inter-allelic or inter-genic clustering with co-regulated genes (*Brickner et al., 2015*; *Brickner et al., 2019*; *Brickner et al., 2016*; *Brickner et al., 2012*; *Kim et al., 2019*; *Kim et al., 2017*; *Randise-Hinchliff et al., 2016*). Loss of nuclear pore proteins or transcription factors that bind to DNA zip codes disrupts clustering (*Brickner et al., 2012*). Clustering has been observed using microscopy as a significant shortening of the distances between two loci in the population (*Brickner et al., 2012*) or using biochemical methods such as 3C/HiC (*Kim et al., 2019*; *Kim et al., 2017*). To explore the dynamics of inter-chromosomal clustering, we tracked the positions and inter-genic distances of well-characterized loci over time in live cells (*Figure 6A*). Both *HIS4* and *INO1* show inter-allelic clustering in diploids. Furthermore, inserting DNA zip codes at *URA3* induces clustering with *HIS4* (*URA3:GCN4BS*; *Randise-Hinchliff et al., 2016*) and *INO1* (*URA3:GRS1*; *Brickner et al., 2012*). The *URA3* gene, which does not undergo inter-chromosomal clustering (*Brickner et al., 2012*), and pairs of randomly selected simulated paths served as negative controls.

Similar to snapshots of populations, the distribution of mean distances from each cell over 40 s (200 × 0.21 s) revealed clustering of *HIS4* with itself as well as inter-genic clustering of *HIS4* with *URA3:GCN4BS* upon histidine starvation (*Figure 6B*). Likewise, *INO1* inter-allelic clustering was observed upon inositol starvation. Mutations in the upstream open reading frames that negatively regulate Gcn4 expression (*uORFmt*; *Mueller et al., 1987*; *Mueller and Hinnebusch, 1986*), led to high-level, constitutive inter-allelic clustering of *HIS4* (*Figure 6B*; *Randise-Hinchliff et al., 2016*), while loss of Nup2 disrupted all clustering (*Figure 6B*). Finally, *URA3*, the simulated nucleoplasmic paths, and the simulated peripheral paths showed no clustering. Thus, NPC- and TF-dependent clustering can be observed over time, and the simulated interaction with the NPC is not sufficient to produce clustering.

We also assessed the stability of clustering over time. The lifetimes of clustering (i.e., time two loci remain within 550 nm) increased from ~5 s for unclustered loci to 20–40 s upon clustering (*Figure 6C*). Similarly, the fraction of the total time points in which clustering was observed reflected

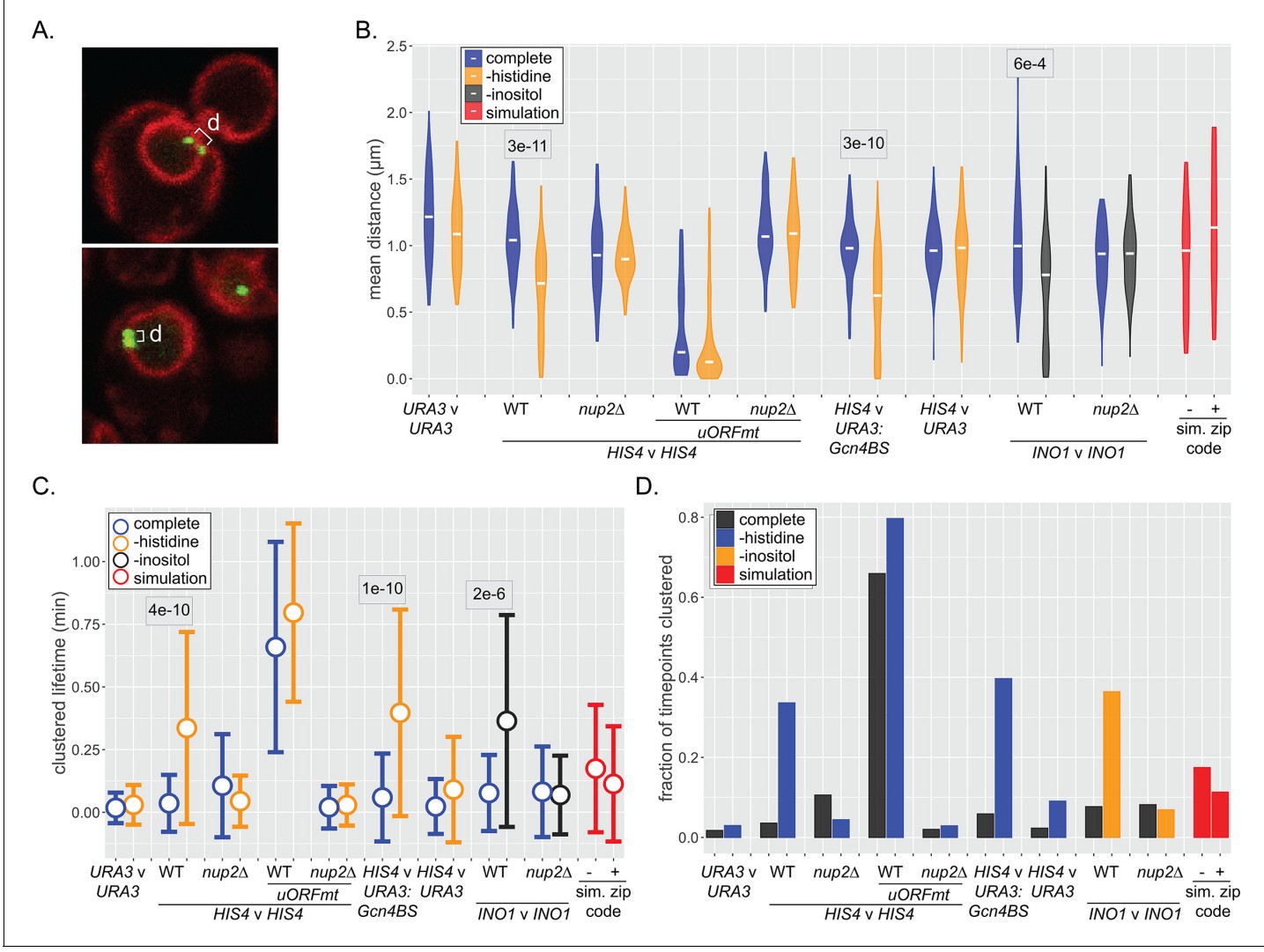

**Figure 6.** Dynamics of inter-chromosomal clustering. (**A**) Confocal micrographs of diploid cells with two loci marked with LacO arrays, expressing LacI-GFP and Pho88-mCherry. Distance between LacO arrays was measured over 200 × 200 ms time points in 40–50 cells (**B–D**). (**B**) Distribution of mean distances between loci for each cell, with the median for each strain or condition indicated with a white dash. p-values<0.05 from the Kolmogorov–Smirnov test are shown. (**C**) Distribution of lifetimes during which d $\leq$ 0.55 µm. Dot = mean, error bars = SD. (**D**) The fraction of all time points that d $\leq$ 0.55 µm for each strain and media condition. For (**B–D**), mean distances, the lifetimes, and fraction of timepoints clustered were also determined for pairs of randomly selected simulated paths (with or without zip code; red).

The online version of this article includes the following source data for figure 6:

**Source data 1.** Comma-separated tables of simulated paths and tracking data used for *Figures 6* and *7*.

the strength of clustering (*Figure 6D*). Because inter-chromosomal clustering persists for relatively long periods of time, it likely reflects a physical interaction.

Finally, we asked if pairs of loci that exhibit clustering show coordinated movement. To quantify the degree of coordination, we determined both the correlation of step sizes by each locus and the average difference in step angles made by each locus over 40 s movies (200 × 0.21 s; *Figure 7A,B*). Uncorrelated movement would result in a correlation of step sizes ~ 0 and a mean difference of angles of ~ π/2 = 1.57 radians for each movie, while perfectly coordinated movement would show a correlation of step sizes ~ 1 and a mean difference of angles ~ 0 (*Figure 7C*). Plotting the correlation and the mean difference in angle for many movies against each other gives a scatter plot (*Figure 7C–L*). As expected, randomly selected pairs of paths generated by the simulation or the simulation+zip code showed no correlated movement (*Figure 7D*). Likewise, nucleoplasmic *URA3*

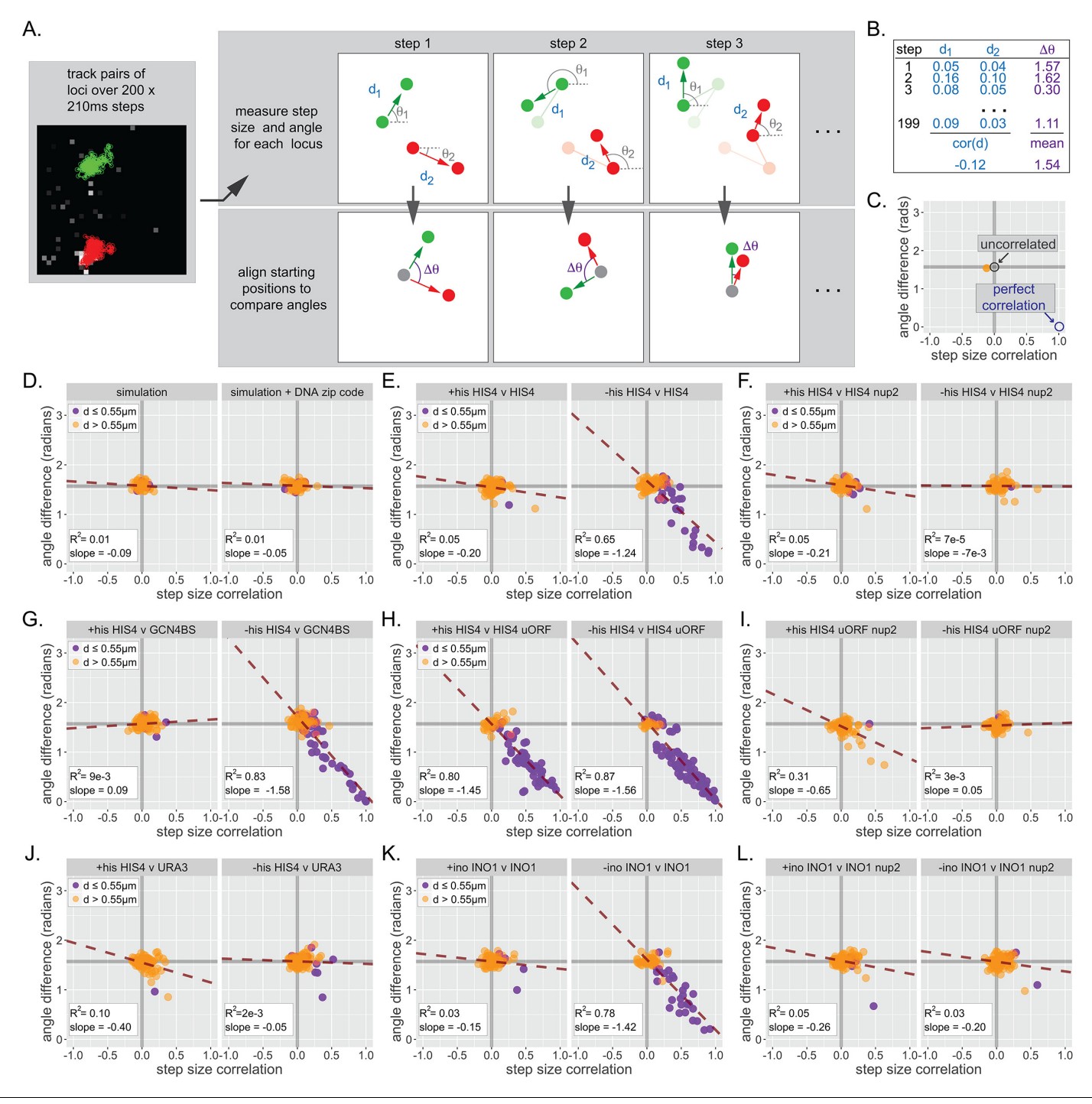

**Figure 7.** Inter-chromosomal clustering leads to coordinated movement. (**A**) Workflow for tracking and analyzing movement of LacO array pairs. For each step from a time series, step distance and step angle are measured (top) and the difference in angles computed (bottom). (**B**) Each time series produces two values: a Pearson correlation coefficient (cor(d)) for all step sizes and a mean difference in angles (Δθ). (**C**) Each cell produces a single point on the summary plot (orange). Gray lines highlight cor(d) = 0 and Δθ = π/2. Uncorrelated movement of two loci would be expected to cluster near cor(d) = 0 and Δθ = π/2, while perfectly correlated movement would result in cor(d) = 1 and Δθ = 0. (**D–L**) Summary plots for correlation analysis of the indicated pairs of loci in the indicated strains grown in the media described in the headers. Cells in which the mean distance between the loci was >0.55 μm appear in orange, while cells in which the mean distance between the loci was ≤0.55 μm appear in purple. For each plot, the slope and $R^2$ for a linear relationship between cor(d) and Δθ are indicated. Forty to 50 cells were analyzed per strain and condition. Simulations are the 50 pairs of paths generated for *Figure 6*.

*Figure 7 continued on next page*

*Figure 7 continued*

The online version of this article includes the following figure supplement(s) for figure 7:

**Figure supplement 1.** Dynamic coordination analysis of *URA3*.

did not show correlated movement with itself (*Figure 7—figure supplement 1*) or with *HIS4* (*Figure 7J*). However, strains that exhibit clustering (i.e., *HIS4* vs *HIS4*, *HIS4* vs *URA3:Gcn4BS*, or *INO1* vs *INO1*) showed a different pattern (*Figure 7E, G, and K*). While the movement of loci that were >0.55 µm (orange dots) apart was uncorrelated, the subset of loci that were ≤0.55 µm (purple dots) showed correlated movement, both in terms of step size and angle. We quantified this behavior using the slope and $R^2$ of the scatter plots (*Figure 7*). Unclustered control loci gave slopes ~ 0 and $R^2 \leq 0.1$ (e.g., *Figure 7D,J*). Under inducing conditions (but not under non-inducing conditions), clustered loci gave a slope closer to the ideal slope of −1.57 and $R^2 \geq 0.65$ (*Figure 7E, G, and K*). Furthermore, overexpression of Gcn4 (*uORFmt*) increased coordinated movement (*Figure 7H*), while loss of Nup2 disrupted coordinated movement (*Figure 7F, I, and L*). Thus, interaction with the NPC, while not sufficient to cause clustering, is required for clustering and coordinated movement. These results indicate that chromosomal loci separated by hundreds of nanometers physically influence each other at a distance.

## Discussion

Tracking yeast NPC-associated chromatin over time revealed a frequent exchange between the nucleoplasm and periphery (*Figure 1*), suggesting that the interaction with the NPC is continuously re-established and that the population averages reflect this dynamism, rather than distinct, stable sub-populations. In other words, localization to the nuclear periphery is less well described as tethering than as a change in the steady-state positioning through continuous binding and dissociation. As interaction with the NPC enhances transcription (*Ahmed et al., 2010*; *Brickner et al., 2019*; *Brickner et al., 2016*; *Brickner et al., 2012*; *Capelson et al., 2010*; *Jacinto et al., 2015*; *Liang et al., 2013*; *Pascual-Garcia et al., 2014*; *Taddei et al., 2006*), it is intriguing that the periodic and transient interaction with the NPC is reminiscent of the widespread phenomenon of transcriptional 'bursting' (*Femino et al., 1998*; *Rodriguez and Larson, 2020*). Transcriptional bursting leads to heterogeneity in the transcription between cells within a population (*Zenklusen et al., 2008*) and disrupting the interaction of the *GAL1-10* promoter with the NPC leads to a decrease in the number of cells expressing these genes without affecting the amount of transcript produced at the site of transcription (*Brickner et al., 2016*). Perhaps interaction with the yeast NPC functions with other transcriptional regulators to stimulate transcriptional bursts. Exploring this connection will require assessing the dynamics of chromatin positioning and transcription simultaneously in live cells.

Chromatin undergoes anomalous sub-diffusive movement during interphase (*Hajjoul et al., 2013*; *Marshall et al., 1997*). The physical interaction between chromatin and the NPC, though transient, reduces chromatin sub-diffusion (*Figure 2*; *Backlund et al., 2014*; *Cabal et al., 2006*), independent of changes in transcription (*Figures 2* and *3*). Using the parameters derived from MSD, we developed computational simulations for yeast chromatin sub-diffusion in the nucleoplasm and at the nuclear periphery. The anticorrelation between successive steps of chromatin and can be modeled as FBM (a.k.a. overdamped fractional Langevin motion; *Lucas et al., 2014*). Sub-diffusion of yeast chromosomal loci is determined by the elastic response from the chromatin polymer and the viscous interaction between the polymer and the nucleoplasm. While we do not explicitly simulate the total chromatin polymer or other nuclear occupants, FBM captures their net effects, recapitulating the MSD behavior of a nucleoplasmic locus (*Figure 4*). However, the FBM model leads to exclusion near boundaries, leading to non-random positioning of loci, a phenomenon that is not consistent with experimental observations. This likely reflects the fact that, while the motion of a segment of chromatin can be modeled as an FBM particle, it is part of a polymer and is not an FBM particle. Our solution to this shortcoming of the FBM model, recalculating the path upon collision with the nuclear boundary (see detailed explanation in Materials and methods), produced localization patterns and MSD behaviors that are consistent with experimental observations. However, additional theoretical and experimental work will help clarify the biological and physical significance of this modification.

To simulate the interaction of chromatin with the NPC, we allowed loci in an area within 50 nm of the nuclear boundary to 'bind' to the nuclear periphery, assuming the mobility of the SPB. The width of this annulus is roughly equal to the height of the NPC nuclear basket (*Vallotton et al., 2019*), whose components are required for chromatin association with the NPC (*Ahmed et al., 2010*). We independently optimized the probability of binding and of remaining bound by comparing the positioning and MSD of simulated paths with that conferred by a DNA zip code. This simple modification of the simulation was able to reliably recreate the peripheral localization and constraint on chromatin sub-diffusion caused by interaction with the NPC (*Figure 4*). Thus, the work described here provides a straightforward and powerful theoretical framework for modeling the biophysical nature of gene positioning through association with any stable nuclear structure.

Repositioning of genes to the NPC during transcriptional activation occurs over a wide range of timescales, depending on the stimulus and gene (*Randise-Hinchliff et al., 2016*), making it difficult to test whether it involves super-diffusive or vectorial movement. Our simulated trajectories offer an important insight; starting from random positions within the center of the yeast nucleus, the population shifted from a random distribution to a peripheral distribution within ~2 min by random sub-diffusion (*Figure 5G*). This timescale is comparable to the experimental models for peripheral repositioning (*Figure 5*), arguing that active mechanism(s) are unnecessary to explain the observed rate of repositioning. More importantly, experimental analysis of the speed and vector of individual steps preceding contact with the nuclear envelope showed non-vectorial sub-diffusive movement that was indistinguishable from that captured by the simulation (*Figure 5*). Furthermore, there was also no difference between experimental cells and negative control cells for these components. These results indicate that zip code-dependent gene localization results from random sub-diffusive chromatin movement, collision with the NPC, leading to dynamic binding. The recently discovered role for actin and Myo3 in localization of *INO1* at the nuclear periphery (*Wang et al., 2020*), raises an important question: how do these factors impact peripheral repositioning through a sub-diffusive mechanism? Our results suggest that loss of Myo3 delays arrival of some loci at the nuclear periphery but does not disrupt localization once it is established (*Figure 5—figure supplement 1*). Perhaps, like actin (*Kapoor et al., 2013*), Myo3 impacts the function of chromatin remodeling complexes or histone-modifying enzymes, which regulate binding of transcription factors to DNA zip codes (*Randise-Hinchliff et al., 2016*). Alternatively, perhaps actin/Myo3 act at the NPC to facilitate capture. A better biochemical and biophysical understanding of these processes will illuminate such possible roles.

Interaction with nuclear pore proteins plays a conserved role in promoting transcription. However, while interaction of yeast genes with nuclear pore proteins occurs at the nuclear periphery in association with the NPC, many genes in mammalian cells and *Drosphila* interact with soluble nuclear pore proteins in the nucleoplasm (*Capelson et al., 2010*; *Liang et al., 2013*; *Light et al., 2013*). Sub-diffusion for mammalian chromatin (which has been suggested to be less mobile than in yeast; *Chubb et al., 2002*) in a nucleus with a radius of 5 μm would make it impossible (on a biologically meaningful timescale) for loci in the center of the nucleus to reach the periphery. In larger nuclei, recruitment of nuclear pore proteins to sites of action, regardless of their position, likely overcomes this obstacle.

Inter-chromosomal clustering is a widespread phenomenon in eukaryotes (*Bantignies et al., 2011*; *Brickner et al., 2012*; *Brown et al., 2008b*; *Cook and Marenduzzo, 2018*; *Eskiw et al., 2010*; *Gehlen et al., 2012*; *Haeusler et al., 2008*; *Noma et al., 2006*; *Ramos et al., 2006*; *Taddei et al., 2009*; *Thompson et al., 2003*; *Xu and Cook, 2008*). Genes that interact with the NPC through shared transcription factors exhibit inter-chromosomal clustering (*Brickner et al., 2015*; *Brickner et al., 2016*; *Brickner et al., 2012*; *Kim et al., 2019*; *Kim et al., 2017*; *Randise-Hinchliff et al., 2016*). Such clustering requires transcription factor(s) and nuclear pore proteins (*Brickner et al., 2012*; *Chowdhary et al., 2017*; *Kim et al., 2019*) but is also mechanistically distinguishable from interaction with the NPC (*Brickner et al., 2016*). Clustering persisted for 20–40 s (*Figure 6*) and led to correlated movement between pairs of loci that were within 550 nm (*Figure 7*). Importantly, independently correlating step size and step angle is sensitive to correlations among pairs of loci in a subset of the cells in the population. Such correlated movement, averaged over the entire population, would be more difficult to appreciate. This may explain why previous work tracking movement of pairs of active *GAL1-10* alleles in yeast found little correlation in aggregate (*Backlund et al., 2014*).

Pairs of paths generated by either the simulation or the simulation+zip code do not lead to inter-chromosomal clustering, consistent with the observation that genes that interact with the NPC through different transcription factors do not exhibit clustering (*Brickner et al., 2012*). Therefore, while clustering requires transcription factors and interaction with the NPC, it represents a distinct physical interaction. Surprisingly, correlated movement was observed between loci separated by hundreds of nanometers, suggesting that it reflects a large molecular complex, or more likely, an environment. Physical interactions that lead to phase separation could encompass groups of genes to create a (perhaps transient) nuclear sub-compartment (*Hult et al., 2017*). This is reminiscent of superenhancers, which exist within phase-separated droplets (*Hnisz et al., 2017*; *Sabari et al., 2018*) and are strongly associated with nuclear pore proteins (*Ibarra et al., 2016*). It is possible that phase separation is facilitated by multivalent interactions between natively unstructured nuclear pore proteins, which are capable of forming phase-separated droplets in vitro (*Frey et al., 2006*; *Frey and Görlich, 2007*). Such conditional phase separation would be regulated and specified by transcription factors, and potentially other transcriptional complexes such as mediator or RNA polymerase II, to functionally compartmentalize the nucleus.

# Materials and methods

## Chemicals, reagents, and media

All chemicals were purchased from Sigma-Aldrich unless otherwise noted. Media components were from Sunrise Science Products, and α-factor was from Zymo Research. Yeast and bacteria media and transformations were as described (*Burke et al., 2000*; *Wood et al., 1983*).

## Yeast strains

All yeast strains were derived from W303 (*ade2-1 ura3-1 trp1-1 his3-11,15 leu2-3,112 can1-100*) strains CRY1 (*MAT*a) or CRY2 (*MAT*α; *Brickner and Fuller, 1997*) and are listed in *Supplementary file 2*. The μNS cytoplasmic particle was expressed from plasmid pAG415GPD-EGFP-μNS (*Munder et al., 2016*).

## Yeast culturing

Yeast cultures were inoculated from a YPD agar plate into synthetic dextrose complete (SDC) or drop out media (*Burke et al., 2000*) and rotated at 30 °C for ≥18 hr, diluting periodically to maintain the cultures at $OD_{600}$ <0.8. Before MSD tracking microscopy, cultures were diluted to ≤0.1 OD/mL and treated with 2 ng/mL of nocodazole for 2 hr. Cultures were then pelleted, washed, and resuspended in SDC to release from M-phase into G1-phase for 10 min. Cells were then pelleted again, concentrated, applied to a microscope slide, and covered with a glass coverslip for imaging.

For experiments involving mating pheromone, 100 μM α-factor was added to the cultures following release from nocodazole arrest for ≥30 min. To release from pheromone arrest, cells were pelleted, washed into SDC, and mounted for microscopy.

## Microscopy

Confocal microscopy was performed in the Northwestern University Biological Imaging Facility. Tracking microscopy was performed on a Leica Spinning Disk Confocal Microscope (Leica DMI6000 inverted microscope equipped with Yokogawa CSU-X1 spinning disk and Photometrics Evolve Delta512 camera), and static localization experiments (*Figures 1B, D*, *3*, and *5C, E*, *Figure 5—figure supplement 1A-C*) were performed on a Leica TCS SP8 Confocal Microscope.

For both single-locus/particle MSD and multiple loci tracking, the same acquisition protocol was used. GFP-LacI/LacO spots in G1-phase cells were imaged every 210 ms for 200 frames in a single *z*-plane with a minimum of 40 biological replicates per experimental condition. Cells that did not remain immobilized or whose loci underwent no movement were excluded from our analysis. For peripheral relocalization dynamics experiments (*Figure 5F, G*, and *I*), LacI-GFP/LacO128 arrays in G1-phase cells were imaged every 500 ms for 600 frames and Pho88-mCherry was imaged every 10 s to determine the position with respect to the nuclear periphery (*D'Urso et al., 2016*; *Egecioglu et al., 2014*).

Static localization experiments (*Figures 1B, D*, *3,* and *5C, E*, *Figure 5—figure supplement 1A-C*) were acquired as *z*-stacks encompassing the full yeast cell, and 30–50 cells were scored per biological replicate as described (*Brickner et al., 2010*; *Brickner and Walter, 2004*; *Egecioglu et al., 2014*). Each strain and condition included at least three biological replicates. To activate light-induced recruitment, cells imaged in *Figure 4C* were scanned with the 488 nm laser every 10 s.

## Particle tracking and data analysis

Tracking was performed using the ImageJ plugin MTrackJ. To accommodate clustering experiments (which typically have two or more fluorescent particles per nucleus), MTrackJ's region of tracking tool was utilized to ensure the signals from individual loci were tracked separately. Tracking data was output as a comma-separated text file and analyzed with R scripts available via GitHub. (https://github.com/MCnu/R_sim_scripts). Repositioning analysis in *Figure 4* utilized a lookup table that contained the frame and the position in which the signal from LacI-GFP/LacO128 array of a given cell first colocalized with the Pho88-mCherry nuclear membrane signal. Tracking data for *Figures 2*, *3*, *5*, *6,* and *7* and simulated paths for *Figures 4*, *5,* and *7* are presented as Source data files associated with each figure.

## FBM simulations

We model the dynamics of chromosomal loci in the cellular nucleus via a discrete-time random walk with continuously varying step sizes. This simulation is governed by FBM, which gives rise to anomalous diffusion of the locus. Anomalous diffusion is distinct from Brownian diffusion due to a non-linear MSD over time, with distinct behaviors for the super-diffusive ($\alpha > 1$) vs. sub-diffusive ($\alpha < 1$) regimes. Free fitting our MSD measurements for 23 different loci/conditions, we found an average $\alpha = 0.52$ (not shown), matching that determined in previous work (*Hajjoul et al., 2013*). Therefore, for the simulations, we used $\alpha = 0.52$. Following previous work (*Lucas et al., 2014*), we present fractional Langevin dynamics simplified by the assumption of overdamping (i.e., no inertial term) and no driving force. In FBM, the statistical noise is a stationary Gaussian process with a mean equal to zero and a nonzero anticorrelation between successive steps (*Meyer et al., 1999*). This property is exploited to allow random vector generation with a given correlation structure (*Dietrich and Newsam, 1997*). We draw values for each simulated dimension of movement to generate the entire time series for a trajectory. We re-scale the vectors to an appropriate magnitude for given time units equal to τ using a Γ parameter provided by non-linear regression on experimental MSD data (where MSD ($\tau$) = $\Gamma(\tau^{0.52})$). No additional complications in our computational model are required to reproduce experimental MSD (*Figure 4—figure supplement 1A,B*).

To properly simulate chromatin diffusion within the confines of the nucleus, we added an impassable boundary to serve as a nuclear membrane. Recent work on the behavior of FBM and the fractional Langevin equation in finite volumes of space showed that the presence of boundaries and the handling of those boundary conditions can affect the long-timescale distribution close to the edges of the domain (*Guggenberger et al., 2019*; *Vojta et al., 2020*; *Vojta et al., 2019*; *Wada and Vojta, 2018*). These studies agree with our findings that in the sub-diffusive regime, depletion occurs at the boundary (*Figure 4—figure supplement 1C, D*). This depletion at the periphery is rationalized by the fact that because successive steps are anticorrelated, a step that would take the particle over the boundary is likely to be followed by one which would take it away from it. Such depletion is not observed in experimental distributions of control and non-control specimens. It is possible that the physicochemical landscape of the periphery or the region near the periphery involves many interactions which have the effect of attracting the chromatin locus to the periphery, but such effects are not evident in the aforementioned studies (which do not consider transient binding interactions with a hard wall). Because our particle is actually a segment of a much larger polymer, we instead decided to regenerate the underlying noise time series whenever the trajectory collides with the periphery to negate the effects of prior movement. This adaptation succeeded in creating a uniform distribution of positions across the nucleus. However, we acknowledge that our theoretical particle no longer satisfies the fluctuation dissipation theorem inherent to all Brownian motion, including FBM. Additional investigation of the behavior of chromatin at the boundary in silica and in vivo will help clarify the validity of this modification.

Binding of chromatin to NPCs was modeled using a simple two-state Markov model wherein a locus within the peripheral region (an annulus extending 50 nm from the nuclear boundary) can assume a bound state in the next step with a defined probability. Particles bound to the NPC remain bound at a second defined probability for every step until it becomes unbound. A particle bound to the NPC is assumed to be interacting strongly with an NPC, their motion is inhibited, but not entirely arrested. We therefore scaled the step sizes of particles in the bound state with $\Gamma$ and $\alpha$ parameters derived from non-linear regression of the MSD for the SPB (*Figure 2*). In this way, we simulate the effective 'pausing' of chromatin motion due to NPC interaction.

## Source code

Our simulation data and source code are openly available. Our simulations were implemented in Python, with routine algorithms like random noise generation or the fast Fourier transform from the NumPy library (*Harris et al., 2020*), and all other codes implemented using custom libraries are available on GitHub (https://github.com/MCnu/YGRW). Analytical pipeline of two-dimensional tracking data is also available. All analyses were implemented in R, and scripts are available on GitHub (https://github.com/MCnu/R_sim_scripts).

## Acknowledgements

The authors would like to thank Dr. Rebecca Menssen and Dr. Madhav Mani for guidance on dynamics analysis; Dr. Reza Vafabakhsh, Dr. Laura Lackner, Dr. Alec Wang, Dr. John Marko, as well as current and former members of the Brickner laboratory for helpful discussions and comments on the manuscript; the Lackner Lab for sharing plasmids, reagents, and guidance with microscopy; the BIF core facility staff at Northwestern University; Dr. Brian Freeman for sharing yeast strains and protocols; Dr. Thomas Vojta for discussions on FBM; and Dr. Yaojun Zhang and Dr. Olga Dudko for access to their MATLAB code used in *Lucas et al., 2014*. MCS was supported by the Cellular and Molecular Basis of Disease NIH T32 GM008061 and SBT received support from the U.S. Department of Energy through the Computational Science Graduate Fellowship under grant number DE-FG02-97ER25308. This work was funded by NIH grants R01 GM118712 and R35 GM136419 and National Cancer Institute U54 CA193419 (JHB).

## Additional information

### Funding

| Funder | Grant reference number | Author |
| --- | --- | --- |
| National Institutes of Health | R01 GM118712 | Michael Chas Sumner<br>Donna G Brickner<br>Jason H Brickner |
| National Institutes of Health | R35 GM136419 | Michael Chas Sumner<br>Donna G Brickner<br>Jason H Brickner |
| National Cancer Institute | U54 CA193419 | Michael Chas Sumner<br>Jason H Brickner |
| National Institutes of Health | T32 GM008061 | Michael Chas Sumner |
| Department of Energy, Labor and Economic Growth | DE-FG02-97ER25308 | Steven B Torrisi |

The funders had no role in study design, data collection and interpretation, or the decision to submit the work for publication.

### Author contributions

Michael Chas Sumner, Conceptualization, Resources, Data curation, Software, Formal analysis, Supervision, Funding acquisition, Investigation, Visualization, Methodology, Writing - original draft, Project administration, Writing - review and editing; Steven B Torrisi, Conceptualization, Software, Formal analysis, Investigation, Visualization, Methodology, Writing - original draft, Writing - review

and editing; Donna G Brickner, Conceptualization, Data curation, Software, Formal analysis, Investigation, Visualization, Methodology, Writing - original draft, Writing - review and editing; Jason H Brickner, Conceptualization, Resources, Software, Formal analysis, Supervision, Funding acquisition, Visualization, Methodology, Writing - original draft, Project administration, Writing - review and editing

### Author ORCIDs
Steven B Torrisi http://orcid.org/0000-0002-4283-8077
Jason H Brickner https://orcid.org/0000-0001-8019-3743

### Decision letter and Author response
Decision letter https://doi.org/10.7554/eLife.66238.sa1
Author response https://doi.org/10.7554/eLife.66238.sa2

## Additional files

### Supplementary files
- Supplementary file 1. Mean squared displacement parameters.
- Supplementary file 2. Strains used in this study.
- Transparent reporting form

### Data availability
All tracking data will be included as Source Data. All Scripts are publicly available from Github https://github.com/MCnu/R_sim_scripts copy archived at https://archive.softwareheritage.org/swh:1:rev:6440995193e1245c44d2c9a9e0b21b161d98e788.

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
