## [Decision Letter]

**Acceptance summary:**

This manuscript combines computational predictions from the physics of motion and experimental live imaging studies to investigate the process of gene re-localization to the nuclear pore complex (NPC) in yeast. The study reveals gene re-positioning to the NPC can be explained by sub-diffusive Brownian motion of loci followed by capture at the NPC rather than through a directed mechanism. The authors also provide new insights into the dynamics underlying the clustering of co-regulated genes. The data and analysis are well-presented and support the authors' conclusions.

**Decision letter after peer review:**

Thank you for submitting your article "Random sub-diffusion and capture of genes by the nuclear pore reduces dynamics and coordinates interchromosomal movement" for consideration by *eLife*. Your article has been reviewed by 3 peer reviewers, one of whom is a member of our Board of Reviewing Editors, and the evaluation has been overseen by Kevin Struhl as the Senior Editor. The following individual involved in review of your submission has agreed to reveal their identity: Kerry Bloom (Reviewer #3).

Essential revisions:

1. The reviewers felt it important to address more explicitly the interplay between the position of the locus within the nucleus and its dynamics. For example, although Figure 2 and 3 show that in activating conditions (-inositol or – histidine), the MSD measurements for INO1 or HIS4 or GCN4 decline (suggesting that nuclear pore contact/tethering leads to reduced movement) this is not further demonstrated more directly by correlating gene position and MSD. Can the authors assess whether loci at the nuclear periphery actually move less than those in the nuclear interior? Although nup2delta mutants in Figure 3 provide further evidence for this, it would be a nice addition to the paper if the relationship between position and movement/MSD were shown more explicitly. This information could likely be gleaned from existing data. An important consideration in this context relates to the resolution of the light microscope and the ability to perceive whether a locus is "at" the periphery. Based on their simulations, can the authors address how this does or does not influence the precision or accuracy of the analysis? The authors decide on 50 nm as their band of "on" and "off" steps in the simulations based on the dimensions of the basket of the NPC, but how are considerations of the microscope resolution accounted for when comparing experiment and theory?

2. On the surface the presented model is somewhat at odds with the recently published report by Wang et al. (Dev Cell 2020, PMID: 31902656), which concludes that peripheral targeting of INO1 depends on a myosin motor and nuclear actin and associated machinery. This should be more directly addressed in the manuscript. Best would be the addition of data in which the authors test the effect of some of these factors in their system and with their measurement. Expansion on this point highlights two other issues for the authors to consider: a) the importance of what can and cannot be gleaned from different approaches (MSD measurements versus population analysis of "on" or "off" the periphery); and b) how a boost in diffusive motion OR directed motion could both influence the ultimate association of a locus with the nuclear periphery. Although the authors present clear evidence against a role for directed motion, the authors also argue against an increase in diffusive motion based on the MSD under conditions that lead to peripheral interactions. However, there does seem to be a trend towards a boost in mobility in an undirected fashion that could still contribute to the kinetics of these processes. Given evidence that transcriptional activation can increase locus mobility from yeast to human cells, it would also extend the study to explore (in simulations) how much of an effect on the kinetics of movement to the periphery would be expected from different extents of such a boost. In other words, could a small (perhaps even insignificant boost based on the errors of the measurements) have a meaningful effect? If not, how much of a boost would be needed? Greater exploration and discussion of these possibilities would be informative for the field.

3. The reviewers questioned whether the data on gene clustering added value to the manuscript, as this aspect of the work was found to be under-developed. The model as applied does not explain the inter-chromosomal clustering (Figure 6). Although informative, how then does binding the NPC dictate clustering? This line of investigation would be made much richer if greater use of modeling could reveal additional parameters that can recapitulate (at least to some extent) the Nup-dependent clustering of genes/alleles. Alternatively, the authors could perhaps further speculate on or test the potential additional forces that drive such clustering. Several studies on clustering chromatin regions indicate these are local interactions (e.g. condensin/transcription factors, Cera et al., 10.1128/mSphere.00063-19 ; Hult et al., doi: 10.1093/nar/gkx741.).

4. The reviewers suggest that the authors expand their discussion and citation of the value for α. Studies from Mikael P. Backlund, Ryan Joyner, Karsten Weis, and W. E. Moerner https://doi.org/10.1091/mbc.e14-06-1127 report the value for α to be 0.6-0.75 in yeast (the same system) and from Weber and Theriot (Weber SC, Spakowitz AJ, Theriot JA (2010) Bacterial chromosomal loci move subdiffusively through a viscoelastic cytoplasm.) to be 0.39 in bacterial systems. This is consistent with the notion that α may realize distinct values at different time scales; the value that they find works well to describe their observations is reasonable but may not always apply to all conditions.

5. Prior work from the Zimmer group argues that there is an effect of the chromosome length in which a locus resides given the impact of the centromeric and telomeric tethers. Do the authors see evidence of this (or in other words, are the loci investigated here on larger chromosomes?). Did the authors consider/explore including such tethers in their simulations?

6. There were several suggestions for greater precision in the language used in the manuscript, including: a) The authors state that zip codes are necessary and sufficient to "cause" peripheral localization, which implies that zip codes drive an active transport mechanism. As the mechanism the authors describe is a random sub-diffusive motion with the zip codes conferring increased dwell time of given loci at the nuclear envelope, stating that zip codes stabilize what would otherwise be very transient interactions would be more appropriate rather than suggesting they direct the localization; b) The reviewers suggest that the authors take greater care with the terms "passive" (p. 14 bottom) and "random". As shown in the Theriot work and many others, these processes are ATP dependent. Motion is severely curtailed upon ATP depletion. It is the case that the motion is random, that does not mean it is passive. In the Introduction the statement that repositioning does not require an active mechanism should also be qualified; active processes contribute to the kinetics and extent of random motion as evidenced by the decrease in MSD in cells depleted for ATP. The authors should clarify that active motion need not equate to a super-diffusive, vectorial mechanism.*Reviewer #1:*

This manuscript explores the mechanisms that govern the movement of gene loci between the nucleoplasm and the nuclear periphery as well as the clustering of gene loci that share common transcriptional activators. The study provides evidence that association of genes with the nuclear pore complex (NPC) or nuclear periphery occurs through a stochastic, constrained diffusion followed by capture that depresses further dynamics of the locus. Clustering of genes under control of a common transcription factor appears to be a distinct process, likely mediated by physical interactions between clustered loci.The authors address the question of whether the behavior of such loci is ergodic – that is, does variation manifest uniformly across the population or are there distinct states in individual cells that contribute, concluding that there is uniform behavior across the population. The MSD analysis and simulations concur on this point and also argue against directed motion. Using a number of genetic tricks they uncover evidence for a constrained diffusion followed by capture model that is unaffected by transcriptional activity. With regards to gene clustering, the investigators argue that such clustering events, although often occurring coincident with recruitment to the nuclear periphery, instead reflects a distinct, likely physical, mechanism.

Overall this is a careful, mechanistically dissected study and the conclusions are well supported by the data. The combination of experimental approaches and simulations is a strength. There are an important inference from these observations, namely that association with the NPC, rather than driving a specific state, instead can contribute to the constraint of a locus in the (larger volume of) nuclear periphery more generally. Another key question in the field, particularly given the broad observation that chromatin dynamics are influenced by the depletion of ATP, has been whether there is a role for actively-driven, directed motion of chromatin to the nuclear periphery and/or NPC – on this point the data is unequivocal, at least for any type of directed motion. The molecular dissection of the contributions to localization and MSD in response to stimuli is impressive and highlights what can be determined in budding yeast relative to other models. In particular, the kinetics of how long it takes for a locus to diffuse and be captured takes is revealing and is supported by experimental observations using the optogenetic approach.

I have some points for the authors to consider further, listed below.

1. The resolution of the light microscope must enter into the ability to perceive whether a locus is "at" the periphery. Based on the simulations, can the authors address how this does or does not enter into the analysis? The authors decide on 50 nm as their band of "on" and "off" steps in the simulations based on the dimensions of the basket of the NPC, but how are considerations of the microscope resolution accounted for when comparing experiment and theory?

2. In Figure 2C what does the line represent a theoretical relationship or the best fit of the data?

3. The authors comment on page 19 about how their observations could be extended to considerations of chromatin movement in the much larger volume of the mammalian nucleus, but putting numbers on this based on their simulation would be very helpful and would extend the study.

4. Prior work from the Zimmer group argues that there is an effect of the chromosome length in which a locus resides given the impact of the centromeric and telomeric tethers. Do the authors see evidence of this (or in other words, are the loci investigated here on larger chromosomes?). Did the authors consider/explore including such tethers in their simulations?

5. In addition to the clear data against a role for directed motion, the authors also argue against an increase in diffusive motion based on the MSD under conditions that lead to peripheral interactions. However, there does seem to be a trend towards a boost in mobility in an undirected fashion that could still contribute to the kinetics of these processes. Given evidence that transcriptional activation can increase locus mobility from yeast to human cells, it would also extend the study to explore (in simulations) how much of an effect on the kinetics of movement to the periphery would be expected from different extents of such a boost. In other words, could a small (perhaps even insignificant boost based on the errors of the measurements) have a meaningful effect? If not, how much of a boost would be needed?

*Reviewer #2:*

In the manuscript by Sumner et al., the authors combine live imaging and computational simulation to investigate the process of gene re-localization to the nuclear pore complex (NPC) in yeast. The authors characterize movement dynamics of genes, previously shown to re-position to the NPC from the nuclear interior upon activation, using the lacO tagging system and mean square displacement (MSD) analysis. They find that genes exhibit transient interactions with the NPC in both induced and uninduced conditions, but that such interactions become more lasting upon activation. Importantly, gene sub-diffusion appears to be reduced by interactions with the NPC, in a way that is not explained by just transcriptional activation.

The authors proceed to successfully model such gene movement and re-positioning, and conclude that gene targeting to the NPC can be explained by sub-diffusive Brownian motion of loci followed by capture at the NPC, as opposed to an active directed mechanism of re-localization. This is an interesting and important conclusion, particularly in light of recently published evidence that nuclear actin and myosin are involved in gene movement to the NPC. The authors' approach also suggests that unlike reported movement of damaged loci, which has been shown to occur in a directed nuclear actin-dependent manner, activated genes re-localize primarily through random movement and collision-capture at the NPC.

On the other hand, the authors' modeling was not found to recapitulate the NPC-mediated inter-chromosomal clustering of genes (previously reported for genes or alleles targeted by the same transcription factor). The authors conclude that gene clustering must involve additional forces not accounted for in their modeling.

Overall, the experiments and analysis appear very rigorous, and the conclusions are impactful, introducing new insight into how active genes re-localize within nuclear space. The presented model will be important for future consideration of other contexts of gene and chromosome movement.

Comments for the authors:

1. Figure 2 and 3 show that in activating conditions (-inositol or – histidine), the MSD measurements for INO1 or HIS4 or GCN4:BS decline, suggesting that nuclear pore contact/tethering leads to reduced movement. This is however not demonstrated more directly though correlating gene position of the gene and MSD. Can the authors assess whether loci at the nuclear periphery actually move less than those in the nuclear interior? Although nup2delta mutants in Figure 3 provide further evidence for this, it would be a nice addition to the paper if the relationship between position and movement/MSD were shown more explicitly. This information may already be in the existing live imaging videos.

2. Since the presented model is somewhat at odds with the recently published report by Wang et al. (Dev Cell 2020, PMID: 31902656) that peripheral targeting of INO1 depends on a myosin motor and nuclear actin and associated machinery, can the authors test the effect of some of these factors in their system and with their measurements? It would be informative for the field to know whether perhaps it is the reduced movement of such genes at the NPC or the capture by the NPC that is primarily affected by the myosin/actin machinery.

3. The title states that "…capture of genes by the nuclear pore.…coordinates interchromosomal movement", yet sub-diffusion and NPC interactions do not seem to explain inter-chromosomal clustering in the presented modeling (Figure 6). Although this is likely not trivial, can the authors attempt to add a modeling parameter that can at least partially reflect the Nup-dependent clustering of genes/alleles? This is certainly not a requirement for publication in my opinion, but it would make this portion of the paper more developed (since currently it reads somewhat underdeveloped). Alternatively, the authors could perhaps further speculate on or test the potential additional forces that drive such clustering.

*Reviewer #3:*

This manuscript follows up work from the investigators' laboratory on the interaction of gene loci with the nuclear envelop. In this manuscript they address the mechanism that drives the interaction and find that it can be explained by sub-diffusive motion, that has been well-characterized at this point for chromosomes in living cells. From the polymer physics perspective, this is entirely consistent with the statistical mechanics of these systems. The work does not preclude findings in the literature that point to active, vectorial processes in specific cases.

The authors state that zip codes are necessary and sufficient to "cause" peripheral localization. This implies the zip codes are part of the transport mechanism. My understanding is that mechanism is a random sub-diffusive motion, and the zip codes increase the dwell time of a given loci to the nuclear envelop. The zip codes don't cause the localization, they stabilize what would otherwise be very transient interactions.

I would caution the authors in citing one value for α. Studies from Mikael P. Backlund, Ryan Joyner, Karsten Weis, and W. E. Moerner https://doi.org/10.1091/mbc.e14-06-1127 report the value for α to be 0.6-0.75 in yeast (the same system) and from Weber and Theriot (Weber SC, Spakowitz AJ, Theriot JA (2010) Bacterial chromosomal loci move subdiffusively through a viscoelastic cytoplasm.) to be 0.39 in bacterial systems.

The authors describe a fractional Brownian model that accounts for the experimental findings. This is not particularly surprising based on several papers in the literature that recapitulate the motion of loci within a chromosome (Verdaasdonk et al., Mol. Cell 2013) and nuclear motion (Arbona et al., Genome Biology, 2017). One needs to be cautious about stating these are "passive" processes (p. 14 bottom). As shown in the Theriot work and many others, these processes are ATP dependent. Motion is severely curtailed upon ATP depletion. It is the case that the motion is random, that does not mean it is passive.

The last figure on clustering raises questions outside the scope of this paper. The authors state that binding the nuclear pore is required for clustering, but it is unclear how binding the pore dictates clustering. Several studies on clustering chromatin regions indicate these are local interactions (e.g. condensin/transcription factors, Cera et al., 10.1128/mSphere.00063-19 ; Hult et al., doi: 10.1093/nar/gkx741.).

I found the data to be very well done and support the authors claims. The suggestions pertain to clarifying several statements and distinguishing between passive processes from random ATP dependent processes.

---

## [Author Response]

Essential revisions:1. The reviewers felt it important to address more explicitly the interplay between the position of the locus within the nucleus and its dynamics. For example, although Figure 2 and 3 show that in activating conditions (-inositol or – histidine), the MSD measurements for INO1 or HIS4 or GCN4 decline (suggesting that nuclear pore contact/tethering leads to reduced movement) this is not further demonstrated more directly by correlating gene position and MSD. Can the authors assess whether loci at the nuclear periphery actually move less than those in the nuclear interior? Although nup2delta mutants in Figure 3 provide further evidence for this, it would be a nice addition to the paper if the relationship between position and movement/MSD were shown more explicitly. This information could likely be gleaned from existing data. An important consideration in this context relates to the resolution of the light microscope and the ability to perceive whether a locus is "at" the periphery. Based on their simulations, can the authors address how this does or does not influence the precision or accuracy of the analysis? The authors decide on 50 nm as their band of "on" and "off" steps in the simulations based on the dimensions of the basket of the NPC, but how are considerations of the microscope resolution accounted for when comparing experiment and theory?

This is an excellent suggestion. First, to clarify our methods, based on our microscopy experiments, we assume that the resolution of our images leads to scoring of a locus as colocalized with the nuclear envelope if it is within 150nm. For our simulations, we assume that the physical interaction with the NPC can only occur if the locus is within 50nm of the nuclear envelope but we score any locus that is within 150nm of the membrane was scored as peripheral. This was described in the manuscript, but we have made it more explicit in multiple locations in the Results section.

Second, we attempted to use our videos to analyze individual steps that occur at the nuclear periphery. This necessitated automating the tracking and gene localization scoring. We were successful in automating the localization scoring and the tracking for these experiments and we validated that the new method produces tracking data very similar to our original method (Figure 2 Supplement 1A). This new tracking method also recapitulated the MSD change upon activation of *INO1* (Figure 2 Supplement 1B). We found that analyzing MSD of individual steps was challenging because individual step sizes are very small, making the measurements more error-prone and the differences less obvious than when examining multiple time intervals. Therefore, to address the reviewers’ question, we exploited the heterogeneity within the population (as demonstrated in Figure 1) to separately analyze the MSD (over multiple time intervals) from cells in which the locus was stably at the nuclear periphery and cells in which the locus was mostly in the nucleoplasm. When such analysis was performed for repressed *INO1*, we observed no difference in MSD between these two populations (Figure 2 Supplement 1C). However, when such analysis was performed for active *INO1*, the cells in which the locus was stably maintained at the nuclear periphery over most of the 40s video showed significantly lower MSD than cells in which the locus was mostly nucleoplasmic (Figure 2 Supplement 1D). This, combined with our genetic experiments showing that loss of Nup2 leads to an increase in MSD for active genes, supports our model that interaction with the NPC is responsible for the drop in sub-diffusion.

2. On the surface the presented model is somewhat at odds with the recently published report by Wang et al. (Dev Cell 2020, PMID: 31902656), which concludes that peripheral targeting of INO1 depends on a myosin motor and nuclear actin and associated machinery. This should be more directly addressed in the manuscript. Best would be the addition of data in which the authors test the effect of some of these factors in their system and with their measurement.

To understand the role of Myo3, we have performed both gene positioning experiments and chromatin mobility experiments in strains lacking Myo3. Our experiments clarify the nature of the defect of *myo3* mutants in repositioning of *INO1* to the nuclear periphery. We find that loss of Myo3 due to either deletion of the gene or degradation of Myo3 using the auxin-inducible degron (AID) slows the rate at which the population shifts to the nuclear periphery upon activation of *INO1* (Figure 5 Supplement 1A and B). Importantly, this defect is specific to one (GRS1) of the two DNA zip codes that mediate repositioning of *INO1* to the nuclear periphery, which we can observe by inserting *INO1* with GRS1 but lacking GRS2 at the *URA3* locus (*URA3:INO1*). When both zip codes (GRS1 and GRS2) are present at the endogenous *INO1* gene, loss of Myo3 had no effect. GRS1-mediated repositioning of *URA3:INO1* to the nuclear periphery occurs after 3-6h, instead of ~1h. Finally, once positioned at the nuclear periphery, *URA3:INO1* localization was unaffected by degradation of Myo3-AID (Figure 5 Supplement 1C).

These results suggest that Myo3 is either impacting the GRS1-dependent movement of *URA3:INO1* from the nucleoplasm to the nuclear periphery or that it is impacting other regulatory steps that are necessary for GRS1-mediated peripheral localization.

We also measured MSD of *URA3:INO1* under repressing, activating (1h) and activating (overnight) conditions. Consistent with delayed repositioning, the drop in MSD observed in the wild type strain after 1h in activating conditions was not observed in the *myo3∆* mutant, while both strains showed the drop after 24h (Figure 5 Supplement 1D).

Because we see no evidence of directed or super-diffusive movement during repositioning to the nuclear periphery (Figure 5), we did not subject the *myo3∆* mutant to these analyses.

Expansion on this point highlights two other issues for the authors to consider: a) the importance of what can and cannot be gleaned from different approaches (MSD measurements versus population analysis of "on" or "off" the periphery); and b) how a boost in diffusive motion OR directed motion could both influence the ultimate association of a locus with the nuclear periphery. Although the authors present clear evidence against a role for directed motion, the authors also argue against an increase in diffusive motion based on the MSD under conditions that lead to peripheral interactions. However, there does seem to be a trend towards a boost in mobility in an undirected fashion that could still contribute to the kinetics of these processes. Given evidence that transcriptional activation can increase locus mobility from yeast to human cells, it would also extend the study to explore (in simulations) how much of an effect on the kinetics of movement to the periphery would be expected from different extents of such a boost. In other words, could a small (perhaps even insignificant boost based on the errors of the measurements) have a meaningful effect? If not, how much of a boost would be needed? Greater exploration and discussion of these possibilities would be informative for the field.

This is an interesting suggestion. We have tested if our observed MSD behavior is compatible with a small increase in chromatin dynamics using our simulations (Author response image 1). Increasing or decreasing the gamma term of the simulations by 10% gives significantly increased and decreased MSD, respectively. Our MSD data would be sensitive to changes in gamma of > 1%. While it is possible that such a change occurs, they may be more apparent at shorter time scales.

We also asked how a 10% increase or decrease in gamma would impact the time of arrival in our simulations. While arrival time is related to gamma, the differences in arrival time produced by increasing or decreasing gamma by 10% are not statistically significant. Therefore, we prefer to avoid including this complication in our model.

3. The reviewers questioned whether the data on gene clustering added value to the manuscript, as this aspect of the work was found to be under-developed. The model as applied does not explain the inter-chromosomal clustering (Figure 6). Although informative, how then does binding the NPC dictate clustering? This line of investigation would be made much richer if greater use of modeling could reveal additional parameters that can recapitulate (at least to some extent) the Nup-dependent clustering of genes/alleles. Alternatively, the authors could perhaps further speculate on or test the potential additional forces that drive such clustering. Several studies on clustering chromatin regions indicate these are local interactions (e.g. condensin/transcription factors, Cera et al., 10.1128/mSphere.00063-19 ; Hult et al., doi: 10.1093/nar/gkx741.).

We agree that future work should explicitly model inter-chromosomal clustering. However, we respectfully disagree that the clustering experiments are under-developed. Using rigorous methods and molecular genetics, this work shows for the first time that NPC-dependent interchromosomal clustering is remarkably stable (Figure 6) and that it leads to coordinated chromosomal movement (Figure 7). This is a valuable contribution to the field and is an important step forward in our ability to model the physical forces responsible for clustering. Furthermore, the fact that our simulations do not produce clustering is itself an interesting and important result, highlighting the idea that interaction with the NPC is necessary but not sufficient to produce clustering and that clustering represents a distinct physical phenomenon.

We are currently working to incorporate the appropriate physical interactions to model this. However, this is a significant undertaking and, we believe, not essential to support the conclusions of the present work.

4. The reviewers suggest that the authors expand their discussion and citation of the value for α. Studies from Mikael P. Backlund, Ryan Joyner, Karsten Weis, and W. E. Moerner https://doi.org/10.1091/mbc.e14-06-1127 report the value for α to be 0.6-0.75 in yeast (the same system) and from Weber and Theriot (Weber SC, Spakowitz AJ, Theriot JA (2010) Bacterial chromosomal loci move subdiffusively through a viscoelastic cytoplasm.) to be 0.39 in bacterial systems. This is consistent with the notion that α may realize distinct values at different time scales; the value that they find works well to describe their observations is reasonable but may not always apply to all conditions.

This is an important caveat to any experiments with mean squared displacement. The value of a that we selected (0.52) was based on the excellent agreement between published work examining many loci in budding yeast (Hajjoul et al. 2013) and the mean a from our MSD experiments on nucleoplasmic loci (Figure 2 and Table S1; described in Methods section). However, we agree that a single a for all chromatin is likely an over-simplification and we have made it clear in the text that we are using it as an average exponent for nucleoplasmic yeast chromatin.

5. Prior work from the Zimmer group argues that there is an effect of the chromosome length in which a locus resides given the impact of the centromeric and telomeric tethers. Do the authors see evidence of this (or in other words, are the loci investigated here on larger chromosomes?). Did the authors consider/explore including such tethers in their simulations?

We explored the effect of tethers on experimental MSD behavior in Figure 2C. We observed an effect of nearby tethering on MSD. Previous work showed that strong effects of tethering are observed only for loci within 30kb of the tether (Avşaroğlu et al., 2014; Verdaasdonk et al., 2013). Because only 15% of the yeast genome is within 30kb of a telomere or a centromere, we did not include this in our simulation of the average nucleoplasmic locus.

6. There were several suggestions for greater precision in the language used in the manuscript, including: a) The authors state that zip codes are necessary and sufficient to "cause" peripheral localization, which implies that zip codes drive an active transport mechanism. As the mechanism the authors describe is a random sub-diffusive motion with the zip codes conferring increased dwell time of given loci at the nuclear envelope, stating that zip codes stabilize what would otherwise be very transient interactions would be more appropriate rather than suggesting they direct the localization; b) The reviewers suggest that the authors take greater care with the terms "passive" (p. 14 bottom) and "random". As shown in the Theriot work and many others, these processes are ATP dependent. Motion is severely curtailed upon ATP depletion. It is the case that the motion is random, that does not mean it is passive. In the Introduction the statement that repositioning does not require an active mechanism should also be qualified; active processes contribute to the kinetics and extent of random motion as evidenced by the decrease in MSD in cells depleted for ATP. The authors should clarify that active motion need not equate to a super-diffusive, vectorial mechanism.

We appreciate the reviewers’ advice on the language in the manuscript. While we do not agree that the word “cause” implies an active, vectorial transport mechanism, we agree that terms such as “target” could be interpreted as such and that the term “passive” is not exactly right. Therefore, we have modified the text to replace “passive” with “random” and the use of the word “targeting” has been replaced with either “repositioning”, “relocalization” or “interaction with the NPC.” Finally, we have taken care to avoid any confusion around the word “cause.”